# LLINBO: TRUSTWORTHY LLM-IN-THE-LOOP BAYESIAN OPTIMIZATION

## ABSTRACT

Bayesian optimization (BO) is a sequential decision-making tool widely used for optimizing expensive black-box functions. Recently, Large Language Models (LLMs) have shown remarkable adaptability in low-data regimes, making them promising tools for black-box optimization by leveraging contextual knowledge to propose high-quality query points. However, relying solely on LLMs as optimization agents introduces risks due to their lack of explicit surrogate modeling and calibrated uncertainty, as well as their inherently opaque internal mechanisms. This structural opacity makes it difficult to characterize or control the exploration–exploitation trade-off, ultimately undermining theoretical tractability and reliability. To address this, we propose LLINBO: LLM-in-the-Loop BO, a hybrid framework for BO that combines LLMs with statistical surrogate experts (e.g., Gaussian Processes ($\mathcal{GP}$)). The core philosophy is to leverage contextual reasoning strengths of LLMs for early exploration, while relying on principled statistical models to guide efficient exploitation. Specifically, we introduce three mechanisms that enable this collaboration and establish their theoretical guarantees. We end the paper with a real-life proof-of-concept in the context of 3D printing.

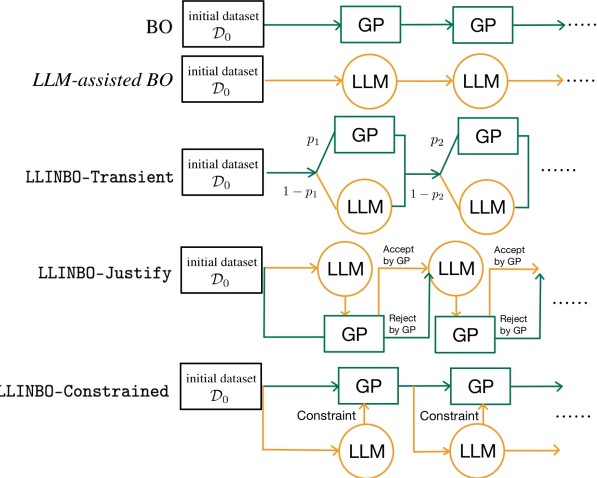

Figure 1: Diagrams of existing methods and the proposed algorithms: `LLINBO-Transient`, `LLINBO-Justify`, and `LLINBO-Constrained`, introduced in Secs. 2.3–2.5.

## 1 INTRODUCTION

BO has emerged as a powerful tool for black-box optimization (BBO), providing a principled framework for balancing exploration and exploitation. BO is particularly useful in scenarios where function evaluations are costly, such as in drug discovery (Korovina et al. (2020)), interaction design (Liao et al. (2023)), and hyperparameter tuning (HPT) (Cho et al. (2020)). Starting with an initial dataset, BO employs a surrogate model, most commonly a $\mathcal{GP}$. The $\mathcal{GP}$ is capable of quantifying uncertainty and is used to approximate both the mean and variance of the black-box function. The next query point, hereafter referred to as a design, is then selected by maximizing an acquisition

function (AF) that quantifies the potential benefit of evaluating a particular point, thereby strategically balancing exploration and exploitation. BO then augments the dataset with the new design–outcome tuple and proceeds sequentially. The past decade has witnessed many success stories for BO, and its theoretical guarantees have been well established for a range of commonly used AFs (Srinivas et al. (2009); Agrawal & Goyal (2012)). These guarantees are typically regret-based, ensuring that, with high probability, one can asymptotically recover an optimal design.

Recently, the few-shot learning capabilities of LLMs and their ability to generate high-quality outputs from minimal examples have made them attractive tools for optimization tasks (Yang et al. (2024)). In particular, LLMs have shown strong empirical performance over random search (Liu et al. (2024)), largely due to their ability to leverage problem context to fast-track the exploration of promising designs. Intuitively, LLMs act like *domain experts*, using contextual cues to identify high-quality designs early in the optimization process. At each iteration, different phases of BO, including initial data generation, proposing new designs, and surrogate modeling, are carried out by the LLM through appropriately tailored prompts (Liu et al. (2024); Yang et al. (2024)). These prompts incorporate the current dataset, typically presented as a list of design-response pairs, together with the problem context, enabling the LLM to function as an optimizer. This prompting framework allows LLMs to act as potential agents for BBO without the need for explicit surrogate modeling or large amounts of observed data. We refer to this class of approaches, where LLMs are solely responsible for proposing design candidates and serve as the surrogate model in BO, as *LLM-assisted BO*.

**Main considerations and contributions.** While recent work on *LLM-assisted BO* (Liu et al. (2024); Guo et al. (2024); Song et al. (2024); Yang et al. (2024)) has demonstrated promise in generating reasonable query designs, several limitations hinder its broader applicability. Most importantly, LLMs do not provide explicit surrogate modeling or calibrated uncertainty, both of which are essential for principled exploration–exploitation trade-offs. Consequently, although LLMs can accelerate optimization in the early stages, their effectiveness systematically degrades as more data are collected and surrogate models strengthen. As we highlight later, this degradation is a central characteristic that we explicitly model and hedge against in our proposed framework.

Moreover, LLMs remain inherently opaque, making the aforementioned trade-off difficult to interpret or control. This structural opacity, combined with their inability to quantify uncertainty in a principled way, introduces significant risks, particularly in applications where cost or safety is critical, ultimately undermining theoretical tractability and reliability. For instance, in the case of smooth functions, the predictive capability of $\mathcal{GP}$s, in terms of both the predicted mean and variance as measured by generalization bounds, has a known rate of improvement as more data is gathered (Srinivas et al. (2009)). The same result is hard to characterize for LLMs, whose internal mechanisms for interpolating black-box functions are not fully understood and which currently lack calibrated uncertainty estimates.

With this in mind, we propose `LLINBO`, a framework that combines the contextual reasoning strengths of LLMs with the principled uncertainty quantification offered by statistical surrogates to enable more trustworthy optimization. To operationalize this collaboration, we introduce a general framework grounded in the philosophy of using LLM-suggested designs to sequentially refine and tailor BO. Within this framework, we propose three approaches, which are inspired by recent developments in federated learning, and analyze the theoretical properties of each. Through extensive simulations and a real-world proof-of-concept in 3D printing, we demonstrate the effectiveness and robustness of the proposed methods.

**Relation to previous works.** LLMs' ability to utilize problem context has been actively investigated. Recent work has also demonstrated that LLMs can generalize effectively from limited in-context information (Lampinen et al. (2025); Brown et al. (2020)), making them particularly promising for BBO, where the objective function is unknown and historical observations are limited (Liu et al. (2024)). The use of LLMs for optimization is a growing research direction. An overview of existing *LLM-assisted BO* can be found in Appendix A.

Based on our best knowledge, the proposed `LLINBO` is the first hybrid framework that integrates both LLMs and $\mathcal{GP}$s into the BO process to accelerate decision-making. We acknowledge that incorporating external information into BO has been investigated in other settings. For example, in Federated BO (F-BO, Dai et al. (2020); Yue et al. (2025); Chen et al. (2025); Dai et al. (2024)),

clients cooperatively perform BO under different sharing schemes. In Human–AI Collaborative BO (HAIC-BO, Hvarfner et al. (2022); Xu et al. (2024); Adachi et al. (2024)), human preferences or belief distributions are incorporated into the BO process. By contrast, the role of LLMs in our framework is fundamentally different from the role of clients in F-BO or humans in HAIC-BO. The few-shot learning ability of LLMs enables the generation of high-quality candidate points in low-data regimes (Liu et al. (2024); Brown et al. (2020)); however, this ability systematically degrades relative to surrogate models as more data accumulate (also demonstrated in our experiments). clients and humans in F-BO and HAIC-BO do not exhibit such properties. This distinction underpins the novelty of our work: `LLINBO` explicitly models this degradation and introduces principled mechanisms to hedge against LLM unreliability while leveraging their early-stage strengths in tandem with $\mathcal{GP}$s.

We also acknowledge that *LLM-assisted BO* is still in its infancy. Existing work primarily focuses on eliciting potentially good designs to evaluate directly from the LLM. This contrasts with BO frameworks that incorporate external guidance, particularly HAIC-BO, where the information elicited from humans is much richer. For instance, $\pi$BO introduced by Hvarfner et al. (2022) requires a preference function from humans, while the method of Xu et al. (2024) relies on an expert function. In comparison, the possibility of eliciting richer forms of information from LLMs beyond a single candidate design per iteration remains largely unexplored. While we see this as an exciting direction for future research, the scope of this paper is on ensuring the safe and trustworthy use of LLM-suggested designs by validating and hedging them with surrogate models.

A detailed review of existing F-BO and HAIC-BO is provided in Appendix A; here, we focus on the works that are most directly relevant to the proposed method. In Dai et al. (2020; 2021), Federated Thompson Sampling for BO was introduced, where clients share $\mathcal{GP}$ Random Fourier Features Rahimi & Recht (2007). Each client then selects the next design to query either based on its own features or on those of another randomly chosen client. Alternatively, Chen et al. (2025) proposed a constraint-sharing strategy, where clients resample their surrogates using shared constraints to guide the next evaluation. While our framework differs in its ultimate objective, these principles have directly inspired our hybrid collaboration between LLMs and statistical surrogates.

## 2 `LLINBO`: LLM-in-the Loop BO

### 2.1 Preliminaries

BO aims to find an optimal design $x^*$ that maximizes a black-box function $f$ over a domain $\mathcal{X}$ by sequentially selecting query designs. Given a total budget of $T$ evaluations, the data at iteration $t \in [T]$ is denoted as $\mathcal{D}_{t-1} = \{(x_i, y_i)\}_{i=1}^{t-1}$, where $y_i = f(x_i) + \epsilon_i$ and $\epsilon_i \sim \mathcal{N}(0, \lambda^2)$.

At time $t$, BO selects the next design, denoted by $x_t$, to observe by maximizing an AF, $\alpha(x, F_{t-1})$, where $F_{t-1}$ is the posterior belief of $f$ conditioned on $\mathcal{D}_{t-1}$. After selecting $x_t$, a noisy observation $y_t = f(x_t) + \epsilon_t$ is obtained, and the dataset is updated as $\mathcal{D}_t = \mathcal{D}_{t-1} \cup \{(x_t, y_t)\}$. This process is then repeated until $T$ is exhausted. The posterior belief is typically modeled using a $\mathcal{GP}$ (Kushner (1964)), which requires a prior mean function $\mu(x)$ (often set to zero) and a kernel function $k(x, x')$ encoding the smoothness of the function. This yields a posterior predictive distribution for $f$ given as

$$f(x) \mid \mathcal{D}_{t-1} \sim \mathcal{GP}(\mu_{t-1}(x), \sigma_{t-1}^2(x)),$$

with $\mu_{t-1}(x) = k_{t-1}(x)^\top (K + \lambda^2 I)^{-1} y$ and $\sigma_{t-1}^2(x) = k(x, x) - k_{t-1}(x)^\top (K + \lambda^2 I)^{-1} k_{t-1}(x)$, where $K$ is the Gram matrix of the training inputs with $K_{ij} = k(x_i, x_j)$, $\forall i, j \in [t-1]$, $k_{t-1}(x) = [k(x, x_1), \ldots, k(x, x_{t-1})]^\top$ being the covariance vector between the input $x$ and the training inputs, and $y = [y_1, \ldots, y_{t-1}]^\top$ is the vector of observed responses.

The posterior mean $\mu_{t-1}(x)$ and variance $\sigma_{t-1}^2(x)$ quantify our posterior belief about the function's value and uncertainty over $\mathcal{X}$, which we denote compactly as $F_{t-1} = \mathcal{GP}(\mathcal{D}_{t-1})$. While many AFs have been proposed and their utility demonstrated, we focus without loss of generality on the Upper Confidence Bound (UCB, Srinivas et al. (2009)), a widely used AF defined as

$$\alpha_{\text{UCB}}(x, F_{t-1}) = \mu_{t-1}(x) + \beta_t \sigma_{t-1}(x), \tag{1}$$

where $\beta_t$ is a parameter that controls the trade-off between exploration and exploitation.

## 2.2 LLM-IN-THE LOOP BO FRAMEWORK

We start by introducing the general framework and define the entity running BO as the client. At each iteration $t$, we assume that the client can prompt an LLM agent $\mathcal{A}$, such as ChatGPT, to suggest a candidate design to query, denoted $x_{\text{LLM},t}$. This interaction can be implemented using a direct prompt from the client to obtain a query design, or through recently developed approaches and prompt templates tailored to the task at hand (Liu et al. (2024; 2025)).Simultaneously, the client learns the posterior belief via a statistical surrogate conditioned on $\mathcal{D}_{t-1}$ and evaluates $x_{\text{LLM},t}$ accordingly. While our framework does not prescribe a specific surrogate model, we assume without loss of generality that the posterior belief is derived from a $\mathcal{GP}$ model, namely, $F_{t-1}$. Specifically, $F_{t-1}$ contains the information of $\mu_{t-1}(x_{\text{LLM},t})$ and $\sigma_{t-1}^2(x_{\text{LLM},t})$, which are used to evaluate $x_{\text{LLM},t}$ with respect to its predicted performance and associated uncertainty. Following this, the client may choose to retain, refine, or reject agent $\mathcal{A}$'s suggestion. For now, we describe this decision step only at a high level in Algorithm 1, as it will be detailed in the three algorithms presented later.

---

**Algorithm 1** LLM-in-the Loop BO Framework (`LLINBO`)

---

**Input:** $\mathcal{D}_0$, $T$, LLM Agent $\mathcal{A}$, kernel function $k$, AF $\alpha$.

 1: **for** $t = 1$ to $T$ **do**
 2:     Compute $F_{t-1} = \mathcal{GP}(\mathcal{D}_{t-1})$
 3:     Compute $x_{\mathcal{GP},t}$ by finding the maximizer of $\alpha(x, F_{t-1})$.
 4:     Query $\mathcal{A}$ for a suggested design point: $x_{\text{LLM},t}$
 5:     Evaluate $x_{\text{LLM},t}$ using $F_{t-1}$
 6:     Generate $x_t$ by refining, retaining or rejecting $x_{\text{LLM},t}$ using mechanisms in Secs. 2.3–2.5
 7:     Obtain $y_t = f(x_t) + \epsilon_t$ and update the dataset: $\mathcal{D}_t \leftarrow \mathcal{D}_{t-1} \cup (x_t, y_t)$
 8: **end for**
 9: **return** $\text{argmax}_{x_i} \{y_i \mid (x_i, y_i) \in \mathcal{D}_T\}$

---

Without steps 4–6 in Algorithm 1, this reduces to BO by selecting $x_t$ as $x_{\mathcal{GP},t}$, and focusing only on step 4 we recover *LLM-assisted BO* approaches, as in Liu et al. (2024). The added steps aim to guide the sampling decision toward more grounded and theoretically justifiable choices that leverage contextual LLM knowledge along with calibrated $\mathcal{GP}$ surrogates and their uncertainty.

We define the instantaneous regret at time $t$ as $r_t = f(x^*) - f(x_t)$, and the cumulative regret as $R_T = \sum_{t=1}^{T} r_t$. The goal is to establish an upper bound on the $R_T$ for all mechanisms to ensure no regret as $T \to \infty$. Our theoretical developments follow the assumptions below:

**Assumption 1.** $f$ belongs to a Reproducing Kernel Hilbert Space (RKHS) $\mathcal{H}_k$ with kernel $k$, such that $\|f\|_{\mathcal{H}_k} \leq B$ for some constant $B \geq 0$ and the kernel satisfies $k(x, x') \leq 1$ for all $x, x' \in \mathcal{X}$. The observational noise $\epsilon_t$ is conditionally $R$-sub-Gaussian for some $R \geq 0$ for all $t \in [T]$.

**Assumption 2.** Let $\gamma_{t-1}$ denote the maximum information gain after time $t - 1$, as defined in Equation (4) of Vakili et al. (2021). AF is defined as in (1), where $\beta_t$ is defined as

$$\beta_t = B + R\sqrt{2(\gamma_{t-1} + 1 + \log\frac{1}{\delta})} \text{ for some } \delta \in (0, 1).$$

## 2.3 LLINBO-TRANSIENT: EXPLORATION BY LLMS THEN EXPLOITATION BY $\mathcal{GP}s$

Perhaps the most natural form of collaboration between an LLM and a BO method is to leverage the LLM's contextual reasoning early in the process, initially placing greater attention on $x_{\text{LLM},t}$, and gradually transition to the $\mathcal{GP}$'s suggestion $x_{\mathcal{GP},t}$ as more data are collected. The $\mathcal{GP}$, with its ability to systematically interpolate observed data and calibrate uncertainty, becomes increasingly reliable for guiding exploitation (Gramacy (2020)).

More specifically, we propose that the query design $x_t$ at iteration $t$ be selected as follows:

$$z_t \sim \text{Bernoulli}(p = p_t), \quad x_t = z_t \cdot x_{\mathcal{GP},t} + (1 - z_t) \cdot x_{\text{LLM},t},$$

where $p_t$ is a monotonically increasing sequence approaching 1 as $t$ increases. Specifically, with probability $p_t$, $x_t$ is set to $x_{\mathcal{GP},t}$, and with probability $1 - p_t$, it is set to $x_{\text{LLM},t}$. The proposed `LLINBO-Transient` algorithm distributes exploration and exploitation across different models:

LLMs facilitate early-stage exploration, while $\mathcal{GP}$s focus on exploitation as more data becomes available. Theoretically, this approach has the following guarantee.

**Theorem 1** (Proof in Appendix B.1). *Suppose that Assumptions 1-2 hold. Let $p_t \in [0, 1]$ be chosen such that $1 - p_t \in \mathcal{O}(1/t)$, Then, with probability at least $1 - \delta$, $R_T$ is upper bounded by*

$$R_T \leq B\mathcal{O}(\sqrt{T}) + \beta_T \mathcal{O}(\sqrt{T\gamma_T}).$$

The assumption on $p_t$ implies that $p_t \to 1$ at rate $1 - \mathcal{O}\left(\frac{1}{t}\right)$. For example, one may choose $p_t = 1 - \frac{1}{t^2}$. With this assumption, the algorithm effectively controls the long-term risk of relying on LLM suggestions throughout the optimization process. Based on this assumption, we apply the Azuma–Hoeffding inequality introduced in Hoeffding (1963) to upper bound the cumulative regret with high probability, which is a standard technique in the BO literature Dai et al. (2020).

## 2.4 LLINBO-JUSTIFY: SURROGATE-DRIVEN REJECTION OF LLM'S SUGGESTIONS

In contrast to the approach in Sec. 2.3, where $x_{\text{LLM},t}$ is directly incorporated during early exploration, here we exploit the posterior believe $F_{t-1}$ as an evaluator for $x_{\text{LLM},t}$. If the LLM suggestion is found to be substantially worse than the current AF maximizer, it is rejected, and $x_{\mathcal{GP},t}$ is used instead. Fundamentally, our goal is to enable the safe integration of LLMs into BO by rejecting suggestions that significantly contradict a client's optimal utility; an approach denoted as LLINBO-Justify.

Specifically, given $x_{\text{LLM},t}$ and the AF constructed by $F_{t-1}$, the client rejects $x_{\text{LLM},t}$ if

$$\alpha_{\text{UCB}}(x_{\text{LLM},t}, F_{t-1}) \leq \max_x \alpha_{\text{UCB}}(x, F_{t-1}) - \psi_t, \tag{2}$$

where $\psi_t$ is the client-selected confidence parameter. The maximum value of the AF, together with the selected $\psi_t$, defines the $\psi_t$-suboptimal region of the AF. Accordingly, $x_{\text{LLM},t}$ is accepted and assigned as $x_t$ if it lies within this region; otherwise, $x_t = x_{\mathcal{GP},t}$.

In the early stages, when the client places greater trust in the LLM's suggestions, a larger $\psi_t$ can be chosen to promote broader exploration around $x_{\text{LLM},t}$, effectively enlarging $\psi_t$-suboptimal region of the AF to investigate a wider area influenced by the LLM. Over time, we recommend gradually decreasing $\psi_t$ to rely more on the $\mathcal{GP}$, whose uncertainty estimates become increasingly well-calibrated as more data is collected. The dynamics of LLINBO-Justify on two benchmark tasks, illustrating how it hedges against poor LLM suggestions, are provided in Appendix C.3.

An upper bound on $R_T$ for LLINBO-Justify is provided in Theorem 2. From (2), we observe that, regardless of whether $x_{\text{LLM},t}$ is accepted or not, the next query design $x_t$ (either $x_{\text{LLM},t}$ or $x_{\mathcal{GP},t}$) always lies within the $\psi_t$-suboptimal region of $\alpha_{\text{UCB}}(x, F_{t-1})$. Leveraging this observation along with classical UCB analysis techniques in Srinivas et al. (2009), the result follows directly.

**Theorem 2** (Proof in Appendix B.2). *Suppose that Assumptions 1-2 hold and $\psi_t \in \mathcal{O}(1/\sqrt{t})$. Then, with probability at least $1 - \delta$, $R_T$ is upper bounded by*

$$R_T = \sum_{t=1}^{T} r_t \leq \sum_{i=1}^{T} \psi_t + 2\beta_T \sum_{i=1}^{T} \sigma_{t-1}(x_t) = \mathcal{O}(\sqrt{T}) + \beta_T \mathcal{O}(\sqrt{T\gamma_T}).$$

## 2.5 LLINBO-CONSTRAINED: CONSTRAIN SURROGATES ON LLM'S SUGGESTIONS

Apart from the two approaches above that depend on defining $p_t$ in LLINBO-Transient and $\psi_t$ in LLINBO-Justify, our third mechanism takes a different approach: it **directly refines the $\mathcal{GP}$** toward potential regions of improvement using $x_{\text{LLM},t}$, without requiring such predefined tuning.

Upon receiving $x_{\text{LLM},t}$, a client treats this as potentially good design. Namely, assumes that $f(x_{\text{LLM},t}) > \kappa_{t-1}$, where $\kappa_{t-1} \triangleq \max_x \mu_{t-1}(x)$ is the posterior mean maximizer. In other words, $x_{\text{LLM},t}$ is treated as a design that can potentially improve upon the current belief of the largest value of $f$. Notice that this constraint may not hold, and we will show shortly how it can be automatically accounted for. With this, the updated posterior belief is given as

$$F_{t-1}^+ \triangleq \mathcal{GP}(\mathcal{D}_{t-1}) \mid \{f(x_{\text{LLM},t}) > \kappa_{t-1}\} \tag{3}$$

This essentially leads to a constrained $\mathcal{GP}$, a $\mathcal{CGP}$. While $\mathcal{CGP}$ does not admit a closed-form posterior, one can readily draw function realizations from it via rejection sampling and approximate the AF using Monte Carlo (MC, Chen et al. (2025)).

In practice, to sample from $F_{t-1}^+$, one can draw $S_t$ realizations, denoted $\tilde{f}_{t-1,s}(x_{\text{LLM},t})$ for $s \in [S_t]$, from $F_{t-1}$. We retain only those samples satisfying the constraint in (3), i.e., $\tilde{f}_{t-1,s}(x_{\text{LLM},t}) > \kappa_{t-1}$. Let $I_t = \{s \mid \tilde{f}_{t-1,s}(x_{\text{LLM},t}) > \kappa_{t-1}\}$ denote the index set of retained samples. For each $s \in I_t$, we construct a $\mathcal{GP}$ based $\mathcal{D}_{t-1} \cup \{(x_{\text{LLM},t}, \tilde{f}_{t-1,s}(x_{\text{LLM},t}))\}$, and denote its posterior mean and variance by $\mu_{t-1,s}^+(x)$ and $\sigma_{t-1,s}^+(x)^2$, respectively.

Fig. 2 illustrates the behavior of `LLINBO-Constrained`. Critically, more output samples are retained when the constraint is satisfied, reflecting posterior support for $x_{\text{LLM},t}$ as a high-quality candidate. In such cases, the mean function under the updated surrogate $F_{t-1}^+$ becomes elevated near $x_{\text{LLM},t}$, highlighting promising regions for subsequent exploration (see Fig. 2(a)–(b)). Conversely, when $x_{\text{LLM},t}$ strongly contradicts the current posterior, no samples are retained ($|I_t| = 0$), and the surrogate remains unchanged, i.e., $F_{t-1} = F_{t-1}^+$, effectively discarding $x_{\text{LLM},t}$ in favor of $x_{\mathcal{GP},t}$ (see Fig. 2(c)–(d)). This selective retention mechanism is key to maintaining the trustworthiness of the BO process and underpins the theoretical guarantees discussed later.

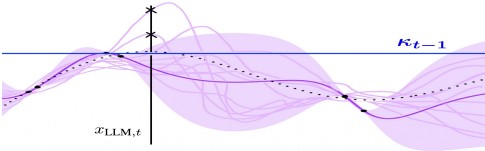

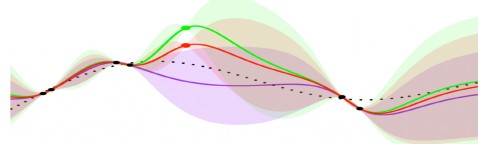

(a) 10 realizations are sampled from $F_{t-1}$ (the light purple curves). Only the points at $x_{\text{LLM},t}$ that are greater than $\kappa_{t-1}$ are retained (the two crosses).

(b) Two $\mathcal{GP}$s (red and green curves and shaded areas) are constructed based on the union of each retained sample and $\mathcal{D}_{t-1}$.

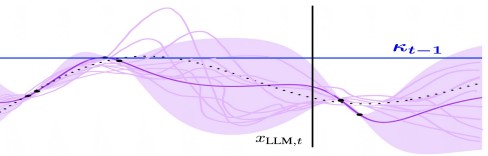

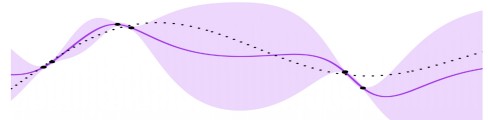

(c) All points lie below than $\kappa_{t-1}$.

(d) The posterior remains unchanged.

Figure 2: Graphical illustration of `LLINBO-Constrained`: solid curve shows $\mathcal{GP}$ mean, shaded area is the confidence interval, and dashed line is the true function $f$.

With these $\mathcal{GP}$s, each constructed from the union of $\mathcal{D}_{t-1}$ and a retained sample, the AF can be approximated via MC methods. Without loss of generality, and focusing on UCB, we can approximate the AF using the law of total variance as

$$\alpha_{\mathcal{CGP}\text{-UCB}}(x, F_{t-1}^+) = \bar{\mu}_{t-1}^+(x) + \tilde{\beta}_t \sqrt{\sigma_{t-1}^+(x)^2 + s_{t-1}^2(x)}, \quad \text{where}$$

$$\bar{\mu}_{t-1}^+(x) = \sum_{s \in I_t} \mu_{t-1,s}^+(x), \quad s_{t-1}(x) = \frac{1}{|I_t| - 1} \sum_{s \in I_t} \left(\mu_{t-1,s}^+(x) - \bar{\mu}_{t-1}^+(x)\right)^2,$$

where $\tilde{\beta}_t$ is the client-specified confidence parameter, which will be discussed in Theorem 3. Notice that the index $s$ is omitted from $\sigma_{t-1,s}^+(x)$ since it is identical for all $s$. This is because the covariance function of a $\mathcal{GP}$ depends only on the input $x$, which is the same across all samples, and not on the sampled responses $\tilde{f}_{t-1,s}(x_{\text{LLM},t})$. Finally, we acquire $x_t$ by solving $x_t = \arg\max_{x \in \mathcal{X}} \alpha_{\mathcal{CGP}\text{-UCB}}(x, F_{t-1}^+)$.

**Theorem 3** (Proof in Appendix B.3). *Suppose Assumption 1 holds. Then, for any $\delta \in (0,1)$ and $T \in \mathbb{N}$, with probability at least $1 - \frac{\delta}{T}$, the following bound holds uniformly for all $t \in [T]$, all retained indices $s \in I_t$, and all inputs $x \in \mathcal{X}$:*

$$\left|\mu_{t-1,s}^+(x) - f(x)\right| \leq \tilde{\beta}_t \sigma_{t-1}^+(x), \text{ where } \tilde{\beta}_t = 2B + 2R\sqrt{2\left(\gamma_t + 1 + \ln\left(\frac{4T}{\delta}\right)\right)} + \sqrt{2\ln\left(\frac{4S_t T}{\delta}\right)}.$$

Theorem 3 includes an additional term involving $S_t$, reflecting the cost of sampling uncertainty. As $S_t$ grows, the potential for deviation increases, requiring a larger $\tilde{\beta}_t$ to maintain the same confidence level. As such, Theorem 3 builds a uniform high-probability bound between the posterior mean of the $\mathcal{CGP}$ and $f$. With this, Theorem 4 then upper bounds $R_T$ for `LLINBO-Constrained`.

**Theorem 4** (Proof in Appendix B.3). *Assume the conditions for Theorem 3 hold and suppose $S_t \in \mathcal{O}(1/t)$. Then, with probability at least $1 - \delta$, $R_T$ satisfies*

$$R_T = \sum_{t=1}^{T} r_t \leq \mathcal{O}\left(\sqrt{T\gamma_T(\gamma_T + \ln(T))}\right).$$

While our theory holds for constant choices of $S_t$, we recommend decreasing $S_t$ as more data is collected, since the surrogate model becomes better calibrated and more reliable over time.

## 3 NUMERICAL STUDIES

We evaluate the proposed methods on two core BO tasks: BBO and HPT, using two representative benchmarks: BO and `LLAMBO`, the most recent state-of-the-art framework introduced by Liu et al. (2024). While effective, implementing `LLAMBO` can be computationally expensive due to the extensive prompting required to generate multiple candidate designs and surrogate evaluations. To mitigate this overhead, we develop `LLAMBO-light`, a lightweight alternative that directly prompts the LLM with the problem context and historical observations to produce the next evaluation design. `LLAMBO-light` serves both as the embedded LLM agent within our proposed three mechanisms and as a baseline. We should note that this is still an emerging area with limited prior work.

For each task with a $D$-dimensional design space, we generate an initial dataset $\mathcal{D}_0$ with $D$ observations. This is done via prompting within the problem context, also known as warmstarting, for methods that utilize LLMs, and via random sampling for BO. To capture the uncertainty in each method's performance, we perform a total of 10 replications. The surrogate model is a $\mathcal{GP}$ with zero prior mean and a Matern kernel. ChatGPT-3.5-Turbo is used as the LLM agent. Detailed implementation of LLM agent, includes structured template and context for each problem can be found in Appendix F. We use UCB as the AF, and set the relevant parameters as follows: $p_t = \min(t^2/T, 1)$, $S_t = 10^4/t^2$, $\psi_t = \frac{1}{t}\sigma_0(x_{\text{LLM},1})$ and $\beta_t = 2\log\frac{tD\pi^2}{0.1*6}$ (as shown effective by Srinivas et al. (2009)).

**BBO task.** We utilize six commonly used simulation functions: Levy-$2D$, Rastrigin-$2D$, Branin-$2D$, Bukin-$2D$, Hartmann-$4D$, and Ackley-$6D$ from Surjanovic & Bingham (2013). For each function, its characteristic patterns and the objective of the problem are incorporated into the prompts as part of the problem context (see Appendix F.1). Performance is reported in terms of the best observed regret, defined as $G_t = f(x^*) - y_t^*$, where $y_t^*$ is the best outcome observed up to time $t$, and $f(x^*)$ denotes the true global maximum. The total budget is set to $T = 10D$.

Fig. 3 shows the regret curves for all methods across the six benchmark functions. Based on these results, we highlight several key insights. First, and perhaps most evidently, `LLAMBO-light` and `LLAMBO` significantly underperform compared to other benchmarks. In many cases, their regret curves remain flat. This supports our motivation: LLMs can assist with BBO but are not yet reliable as standalone agents. Second, methods involving LLMs, including ours, achieve a strong early lead. This suggests that LLMs can effectively leverage problem context to quickly identify promising regions, making them a useful complement to BO frameworks. Third, we observe that our hybrid mechanisms consistently outperform the benchmarks across all functions. This superiority is especially evident in the early rounds and gradually diminishes as more data is collected. This trend is not surprising; statistical surrogate models become more accurate with more data, aligning with our core philosophy of reducing reliance on LLMs as the optimization process progresses.

**HPT task.** We consider two physical simulation functions: the piston (Kenett & Zacks (1998)) and robot arm (An & Owen (2001)), along with three regression models: Random Forest (RF-$4D$), Support Vector Regression (SVR-$3D$), and XGBoost (XGB-$4D$). The total budget is set to $T = 5D$. For each simulation function, we generate 1,000 data points and define the regret as the best-observed Mean Squared Error (MSE) at each iteration, where the MSE is obtained by fitting the corresponding regression model and evaluating it via 10-fold cross-validation. A detailed description of each

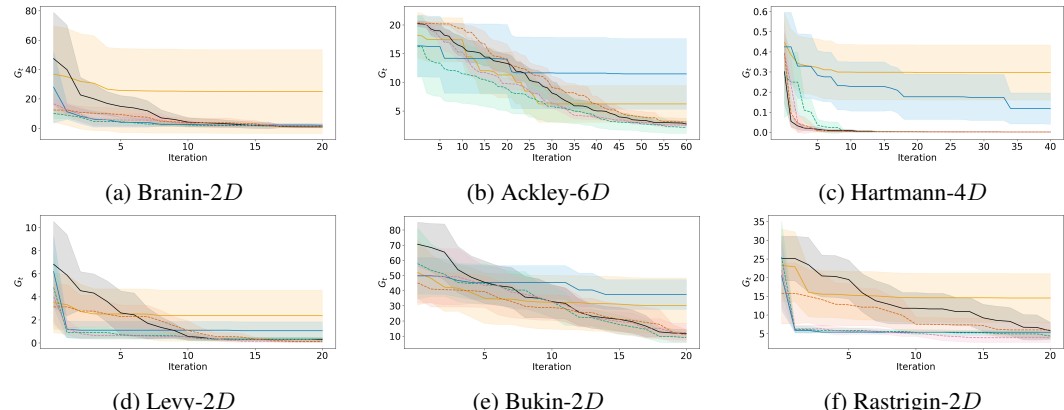

(a) Branin-2$D$     (b) Ackley-6$D$     (c) Hartmann-4$D$

(d) Levy-2$D$     (e) Bukin-2$D$     (f) Rastrigin-2$D$

Figure 3: $G_t$ comparison for BBO. Each line shows the mean regret, shaded with 95% confidence intervals. **Proposed methods**: `--- LLINBO-Transient`, `--- LLINBO-Justify`, `--- LLINBO-Constrained`. **Baselines**: `—— LLAMBO`, `—— LLAMBO-light`, `—— BO`.

data–regression model pair, along with the corresponding problem formulation, is provided in the prompt (see Appendix F.2). The results in Fig. 4 once again confirm the insights from the BBO task. Namely, we find that LLMs are often capable of generating high-quality designs in the early iterations by leveraging the problem context. Furthermore, our proposed LLM-$\mathcal{GP}$ collaborative mechanisms yield significantly lower MSE compared to all benchmarks across the tasks.

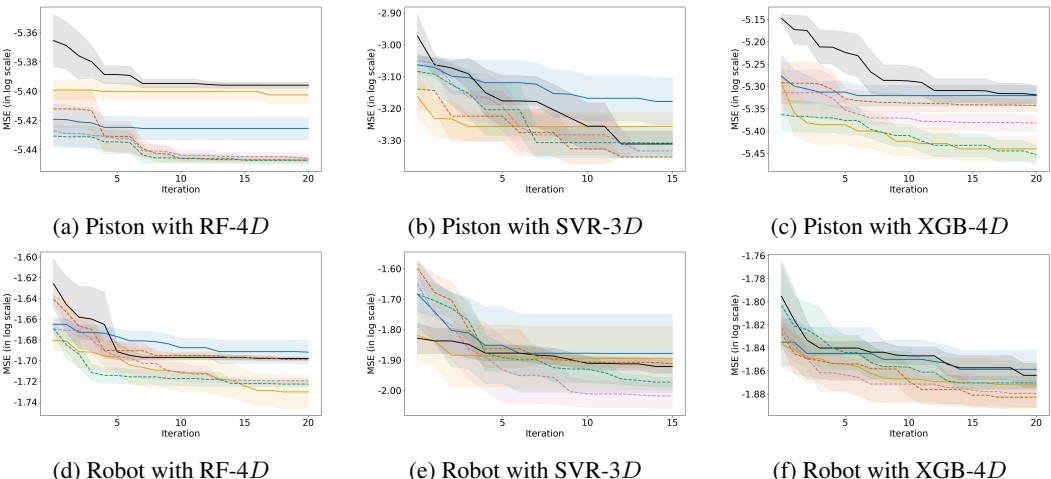

(a) Piston with RF-4$D$     (b) Piston with SVR-3$D$     (c) Piston with XGB-4$D$

(d) Robot with RF-4$D$     (e) Robot with SVR-3$D$     (f) Robot with XGB-4$D$

Figure 4: MSE comparison for HPT. Each line shows the mean MSE, shaded with 95% confidence intervals. **Proposed methods**: `--- LLINBO-Transient`, `--- LLINBO-Justify`, `--- LLINBO-Constrained`. **Baselines**: `—— LLAMBO`, `—— LLAMBO-light`, `—— BO`.

We end by noting that, as highlighted earlier, HAIC-BO methods generally require much richer forms of information from humans than what is elicited from LLMs in *LLM-assisted BO* approaches. This makes a direct comparison between our method and HAIC-BO particularly challenging, even if one were to treat humans as LLMs. Nevertheless, in Appendix C.1 we adapt $\pi$BO introduced by Hvarfner et al. (2022)) so that the belief functions originally provided by humans can instead be extracted from LLMs, and we present a comparison between $\pi$BO and our proposed method.

While all proposed methods perform well across both BBO and HPT tasks, the choice among them ultimately depends on practical requirements. Guidelines for selecting among the variants and tuning their hyperparameters are provided in Appendix D. In Appendices C.2 and C.4, we further examine the performance of the proposed approaches in high-dimensional settings and under different LLM configurations. Finally, Appendix E presents a detailed analysis of their computational complexity.

## 4 APPLICATION TO 3D PRINTING

In addition to the numerical evaluation, we further assess the performance of our method through a case study in 3D printing, aimed at reducing stringing in a printed product. Stringing (Fig. 5(b)) is a prevalent defect in fused filament fabrication (FFF) 3D printing. FFF is commonly used for rapid prototyping and low-cost part production. However, stringing degrades surface quality and often requires additional post-processing (Paraskevoudis et al. (2020)). This study aims to optimize the design parameters of a Creality Ender 3 desktop FFF printer (Fig. 5(a)), including nozzle temperature, Z hop height, retraction distance, outer wall wipe distance, and coasting volume, using stringing percentage as the outcome variable. Further details about the parameters can be found in Appendix G.

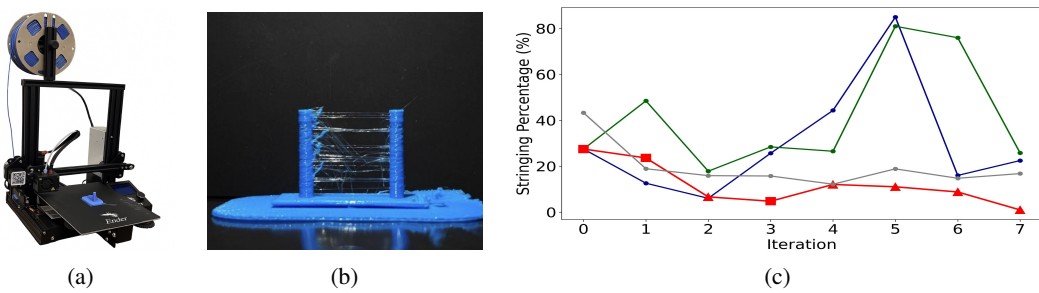

| (a) | (b) | (c) |

Figure 5: Demonstration of 3D printing experiments and results. **(a):** printer used, **(b):** stringing between two columns, **(c):** benchmark results. Benchmarks: — `LLAMBO-light`, — `LLAMBO`, — `LLINBO-Transient`, and — `BO`. For `LLINBO-Transient`, we use square and triangle markers to indicate updates chosen based on an LLM or $\mathcal{GP}$, respectively.

**Experiment setup.** All experiments were conducted on a single printer using PETG filament (Holcomb et al. (2022)), selected for its high tendency to produce stringing (see Fig.5(b)). We adopted a standard two-column geometry with a horizontal gap, commonly used in stringing evaluations (Haque (2020)). At each iteration, after printing the object with the proposed parameters, the stringing percentage (ranging from 0 to 100%) was quantified (details in Appendix G.1).

Due to the cost associated with this experiment (each run takes several hours), we limit our comparison to `LLINBO-Transient` with $p_t = 1 - \frac{1}{t}$, evaluated against `LLAMBO`, `LLAMBO-light`, and BO. All other settings follow Sec. 3. The prompts specifying the problem context and controllable parameters are provided in Appendix G.2. Unlike Sec. 3, the objective here is not full evaluation, but to demonstrate the effectiveness of our method and the broader potential of LLMs in optimal design.

Several insights can be draw from the results shown in Fig. 5(c): (i) Our approach demonstrates strong overall performance and ultimately achieves near-zero stringing. (ii) Methods utilizing LLMs achieve a strong head start compared to BO, highlighting the value of LLMs in optimal design. (iii) Consistent with our simulation results, `LLAMBO` and `LLAMBO-light` perform poorly and do not exhibit a decreasing trend in regret. (iv) While BO shows improvement over time, our hybrid approach outperforms it. This again highlights the collaboration benefits between LLMs and surrogate experts.

## 5 CONCLUSION

The proposed `LLINBO` framework leverages LLMs' contextual reasoning to generate high-quality designs early, while surrogate models refine and guide the search as data accumulates. The mechanisms developed under `LLINBO` exhibit strong performance, as demonstrated by both simulation and real-world case studies. While the use of LLMs in optimization remains in its infancy, we believe this line of research holds great promise for enabling more adaptive, data-efficient, and practical optimization strategies across a wide range of applications. The strength of our hybrid framework depends on parameters that are sensitive to how well the LLM understands the problem context in early stages. A promising direction is to link these parameters to a metric that quantifies an LLM understanding. Our overarching framework can potentially help design LLM-assisted optimization beyond black-box settings.

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

## A  MORE RELATED WORKS

**LLM-assisted BO.**   Recently, with the few-shot learning ability of LLMs to generate high-quality answers from limited input, leveraging LLMs in the BO process has emerged as a promising yet relatively new research direction. For example, Liu et al. (2025) employed LLMs to solve multi-objective optimization problems, while Guo et al. (2024) extended their use to a broader set of tasks, including combinatorial optimization. More recently, Song et al. (2024) explored how LLMs can enhance BBO by leveraging textual knowledge and sequence modeling to improve generalization. SLLMBO is proposed to solve HPT task by combining TPE and the reasoning strength of LLMs (Mahammadli & Ertekin (2022)). A detailed investigation of Kristiadi et al. (2024) is conducted to assess the LLM's ability to assist BO process. These works highlight the potential of LLMs in various optimization settings, a direction that remains actively under investigation.

Recently, a variety of approaches have emerged that leverage LLMs to address black-box optimization. For example, Li et al. (2025) introduced LLaMEA-BO, where an LLM generates and iteratively refines BO pseudocode. FunBO Aglietti et al. (2024) learns novel acquisition functions, represented as Python programs, using FunSearch and achieves improved performance in both in-distribution and out-of-distribution settings. BioDiscoveryAgent Roohani et al. (2024) is an LLM-driven closed-loop system for designing genetic perturbation experiments, outperforming BO baselines by leveraging biological reasoning and tool-augmented analysis.

While recent *LLM-assisted BO* methods involve leveraging LLMs at various stages of the optimization pipeline and across diverse applications, the primary contribution of this paper is not to introduce yet another LLM-based optimizer, but rather to ensure that the optimization process involving an LLM agent is both efficient and trustworthy.

**F-BO**   To enable collaboration between LLMs and statistical surrogates in enhancing BO, while leveraging the distinctive ability of LLMs to provide a set of designs, we draw inspiration from the literature on F-BO. Federated learning (FL) aims to establish a collaborative framework that allows clients to work together while keeping their own data private. This setting has directly influenced prior work in F-BO, where a single design is often shared across clients.

For example, Yue et al. (2025) developed a consensus framework for collaborative BO, where the next design to query is selected as a weighted combination, dictated by a dynamically coupled stochastic consensus matrix, of the AF maximizers from all clients in the system, including each client's own. Other works like Chen et al. (2025) and Dai et al. (2020) require only the design from other clients, while the former requires a design point from other clients directly, and the latter requires Random Fourier Features (Rahimi & Recht (2007)). A recent review on federated and collaborative BO can be found in Al Kontar (2024).

Table 1: Comparison of `LLINBO` with related BO frameworks incorporating external information.

| Feature | `LLINBO` | **HAIC-BO** | **F-BO** | *LLM-assisted BO* |
|---|---|---|---|---|
| Minimal assumption on external info | ✓ | ✗ | ✓ | ✓/✗ |
| Theoretical guarantees | ✓ | ✓/✗ | ✓ | ✗ |
| Handles early LLM strength, later weakness | ✓ | ✗ | ✗ | ✓/✗ |
| Preserves BO structure | ✓ | ✓ | ✓ | ✗ |
| Dynamic reliance adjustment | ✓ | ✗ | ✗ | ✗ |

**HAIC-BO**   In contrast, HAIC-BO requires richer information compared with F-BO, as privacy concerns are not considered in this setting. For instance, COBOL (Xu et al. (2024)) requires explicit beliefs about the function from the user, while CoExBO (Adachi et al. (2024)) relies on preference pairs provided by a human. Similarly, $\pi$BO (Hvarfner et al. (2022)) assumes access to a formal prior distribution specified by a human expert, and the method in (AV et al. (2022)) requires information about good and bad regions of the design space. Another key difference between our framework and HAIC-BO lies in the assumptions placed on LLMs or humans. Notably, our method does not impose any assumptions on the capacity of LLMs; instead, it hedges against poor suggestions through three

distinct hedging processes. By contrast, HAIC-BO often introduces behavioral assumptions about humans; for example, AV et al. (2022) assumes that a human expert follows a BO-like strategy.

We acknowledge all works that tried to cooperate with outside information to make decision-making more efficient, and we use the table below to compare and highlight the key differences between the proposed LLINBO framework and the rich existing works, including HAIC-BO, F-BO, and *LLM-assisted* BO.

## B  TECHNICAL RESULTS

We first introduce two Lemmas that are quite common in BO analysis. Lemma 1 derives the concentration between the posterior mean and the ground truth.

**Lemma 1.** *(Theorem 2 of Chowdhury & Gopalan (2017)) Under Assumption 1 and 2, and let* $\hat{\lambda}_t = 1 + 2/t$. *For arbitrary* $\delta \in (0,1)$*, with probability at least* $1 - \delta$*, we have:*

$$
\begin{aligned}
|\mu_{t-1}(x) - f(x)| &\leq |k_{t-1}(x)^\top (K_{t-1} + \hat{\lambda}_t I)^{-1} [\delta_1, ..., \delta_{t-1}]^\top| \\
&\quad + |f(x) - k_{n,t}(x)^\top (K_{t-1} + \hat{\lambda}_t I)^{-1} [f(x_1), ..., f(x_{t-1})]^\top| \\
&\leq (B + R\sqrt{2(\gamma_{t-1} + 1 + \ln(1/\delta))})\sigma_{t-1}(x) \\
&= \beta_t \sigma_{t-1}(x),
\end{aligned}
\tag{4}
$$

$$
\tag{5}
$$

*where* $\delta_i = f(x_i) - y_i \; \forall i \in [t-1]$.

With this Lemma, we can bound the regret raised at every iteration, which is stated in Lemma 2.

**Lemma 2** (Theorem 3 in Chowdhury & Gopalan (2017)). *Assume that Assumptions 1 and 2 hold. UCB is used to select* $x_t \; \forall t \in [T]$*. With probability at least* $1 - \delta$*, where* $\delta \in (0,1)$*, the regret at time* $t$ *can be upper bounded by*

$$
r_t = f(x^*) - f(x_t) \leq \beta_t \sigma_{t-1}(x_t) + \mu_{t-1}(x_t) - f(x_t) \leq 2\beta_t \sigma_{t-1}(x_t).
$$

Next, when using the UCB as the AF, we present a commonly used lemma that bounds the cumulative posterior variance at the selected design points in terms of the information gain.

**Lemma 3** (Lemma 4 in Appendix of Chowdhury & Gopalan (2017)). *Let* $x_1, \ldots, x_T$ *be the designs selected by the algorithm. Then, the sum of the predictive standard deviations at these points can be bounded by*

$$
\sum_{t=1}^{T} \sigma_{t-1}(x_t) \leq \sqrt{4(T+2)\gamma_T} = \mathcal{O}(\sqrt{T\gamma_T}).
$$

### B.1  PROOF OF THEOREM 1

The proof builds on the approach of Dai et al. (2020), which uses the Azuma-Hoeffding inequality to derive a high-probability upper bound on the regret, transforming the expected regret into a probabilistic guarantee. Recall that when LLINBO-Transient is applied, $x_t$ is selected as

$$
x_t = \begin{cases} x_{\text{LLM},t} & \text{with probability } 1 - p_t \\ x_{\mathcal{GP},t} & \text{with probability } p_t \end{cases}.
$$

Let $A_t$ and $B_t$ be the event when $x_t$ is selected the same as $x_{\text{LLM},t}$ and $x_{\mathcal{GP},t}$, respectively. When event $A_t$ happens, the regret conditioned on $A_t$ can be upper bounded with high probability via Lemma 2. In this case, the expected regret at time $t$ can be controlled via Lemma 4.

**Lemma 4.** *Pick* $\delta \in (0,1)$*, let* $\delta' = \frac{\delta}{2}$ *and define* $\beta_t$ *the same as Assumption 2. Then, with probability at least* $1 - \delta'$*, we have*

$$
\mathbb{E}[r_t | \mathcal{F}_{t-1}] \leq p_t (2\beta_t \sigma_{t-1}(x_{\mathcal{GP},t})) + (1 - p_t)\nu_t,
$$

*where* $\mathcal{F}_{t-1}$ *denotes the filtration until* $t - 1$ *and* $\nu_t = \mathbb{E}[r_t | \mathcal{F}_{t-1}, B_t]$.

*Proof.* As the choice of the next evaluation design is stochastic, one needs to consider the expected regret given the current filter $\mathcal{F}_{t-1}$, which can be written as

$$\mathbb{E}[r_t|\mathcal{F}_{t-1}] = p(A_t)\mathbb{E}[r_t|\mathcal{F}_{t-1}, A_t] + p(B_t)\mathbb{E}[r_t|\mathcal{F}_{t-1}, B_t].$$

Note that the term $\mathbb{E}[r_t|\mathcal{F}_{t-1}, A_t]$ is deterministic and can be upper bounded with probability $1 - \delta'$ via Lemma 2. Let $\nu_t = \mathbb{E}[r_t|\mathcal{F}_{t-1}, B_t]$, we have

$$\begin{aligned}
\mathbb{E}[r_t|\mathcal{F}_{t-1}] &= p_t(f(x^*) - f(x_{\mathcal{GP},t})) + (1 - p_t)\nu_t \\
&\leq p_t(2\beta_t\sigma_{t-1}(x_{\mathcal{GP},t})) + (1 - p_t)\nu_t.
\end{aligned} \tag{6}$$

$\square$

The following lemma is used to transform the expected regret to an unexpected form with high probability.

**Lemma 5.** *(Azuma-Hoeffding Inequality) Given $\delta \in (0,1)$ and a super-martingale $Y_t$, $t \in [T]$. Suppose with probability $1 - \delta$, $Y_t - Y_{t-1} \leq k_t \ \forall t \in [T]$ we have*

$$p\left(|Y_T - Y_0| \leq \sqrt{-2log\delta \sum_{t=1}^{T} k_t^2}\right) > 1 - \delta.$$

Let $X_t = r_t - (p_t(2\beta_t\sigma_{t-1}(x_{\mathcal{GP},t})) + (1 - p_t)\nu_t)$, and define $Y_t = \sum_{s=1}^{t} X_s$ with $Y_0 = 0$. We claim that $Y_t$ forms a super-martingale and hence apply Lemma 5 to bound $Y_T - Y_0 = Y_T$. To verify the super-martingale property of $Y_t$, we compute the conditional expectation of its increments:

$$\begin{aligned}
\mathbb{E}[Y_t - Y_{t-1}|\mathcal{F}_{t-1}] &= \mathbb{E}[X_t|\mathcal{F}_{t-1}] \\
&= \mathbb{E}[r_t - (p_t(2\beta_t\sigma_{t-1}(x_{\mathcal{GP},t})) + (1 - p_t)\nu_t)|\mathcal{F}_{t-1}] \\
&= \mathbb{E}[r_t|\mathcal{F}_{t-1}] - (p_t(2\beta_t\sigma_{t-1}(x_{\mathcal{GP},t})) + (1 - p_t)\nu_t) \\
&\leq 0. \qquad\qquad \text{(by (6))}
\end{aligned}$$

In this case, $Y_t$ is a super-martingale. Next, we derive the upper bound of $|Y_t - Y_{t-1}|$, which is essential for applying Lemma 5:

$$\begin{aligned}
|Y_t - Y_{t-1}| &= |X_t| \\
&= |r_t - (p_t(2\beta_t\sigma_{t-1}(x_{\mathcal{GP},t})) + (1 - p_t)\nu_t)| \\
&\leq |r_t| + p_t(2\beta_t\sigma_{t-1}(x_{\mathcal{GP},t})) + (1 - p_t)\nu_t \qquad \text{(by triangle inequality)} \\
&\leq B + p_t(2\beta_t\sigma_{t-1}(x_{\mathcal{GP},t})) + (1 - p_t)B. \qquad \text{(by Assumption 1)}
\end{aligned}$$

As a result, by Lemma 5 and with probability $1 - \delta'$, $\delta' = \frac{\delta}{2}$,

$$Y_T \leq \sqrt{-2\log\delta' \sum_{t=1}^{T}\left(B + (1 - p_t)B + 2p_t\beta_t\sigma_{t-1}(x_{\mathcal{GP},t})\right)^2}.$$

With some simple algebra and with probability $1 - \delta' - \delta' = 1 - \delta$, we can upper bound the cumulative regret as

$$R_T = \sum_{t=1}^{T} r_t$$

$$\leq \underbrace{\sum_{t=1}^{T} p_t(2\beta_t \sigma_{t-1}(x_{\mathcal{GP},t}))}_{A} + \underbrace{\sum_{t=1}^{T} (1 - p_t)\nu_t}_{B}$$

$$+ \underbrace{\sqrt{-2\log\delta' \sum_{t=1}^{T} (B + (1-p_t)B + 2p_t\beta_t\sigma_{t-1}(x_{\mathcal{GP},t}))^2}}_{C}$$

$$\leq \underbrace{\beta_T \mathcal{O}(\sqrt{T\gamma_T})}_{A} + \underbrace{B\mathcal{O}(\log T)}_{B} + \underbrace{B\mathcal{O}(\sqrt{T}) + B\mathcal{O}(\log T) + \beta_T \mathcal{O}(\sqrt{T\gamma_T})}_{C} \quad \text{(by Lemma 3)}$$

$$= B\mathcal{O}(\sqrt{T}) + \beta_T \mathcal{O}(\sqrt{T\gamma_T}).$$

## B.2 PROOF OF THEOREM 2

The process of selecting $x_t$ via LLINBO-Justify can be written as

$$x_t = \begin{cases} x_{\mathcal{GP},t} \text{ if } \alpha_{\text{UCB}}(x_{\text{LLM},t}, F_{t-1}) < \alpha_{\text{UCB}}(x_{\mathcal{GP},t}, F_{t-1}) - \psi_t \\ x_{\text{LLM},t} \text{ else} \end{cases}.$$

Note that no matter which cases is fulfilled, $x_t$ is the $\psi_t$-suboptimal of $\alpha_{\text{UCB}}(\cdot, \cdot)$. Also, for $\delta \in (0, 1)$ and $\beta_t$ is selected the same as in Assumption 2. We can upper bound $r_t$ by

$$r_t = f(x^*) - f(x_t)$$

$$\leq \underbrace{\mu_{t-1}(x^*) + \beta_t\sigma_{t-1}(x^*)}_{A} - \underbrace{f(x_t)}_{B} \quad \text{(by Lemma 1)}$$

$$\leq \underbrace{\mu_{t-1}(x_{\mathcal{GP},t}) + \beta_t\sigma_{t-1}(x_{\mathcal{GP},t})}_{A} - \underbrace{(\mu_{t-1}(x_t) - \beta_t\sigma_{t-1}(x_t))}_{B} \quad \text{(by Lemma 1)}$$

$$\leq \underbrace{\mu_{t-1}(x_t) + \beta_t\sigma_{t-1}(x_t) + \psi_t}_{A} - \underbrace{(\mu_{t-1}(x_t) - \beta_t\sigma_{t-1}(x_t))}_{B}$$

$$\leq \psi_t + 2\beta_t\sigma_{t-1}(x_t).$$

By assuming that $\psi_t = \mathcal{O}(1/\sqrt{t})$ and by the Lemma 4 in Chowdhury & Gopalan (2017), which allows us to bound the sum of variance at the evaluated designs, we have

$$R_T = \sum_{t=1}^{T} r_t \leq \sum_{i=1}^{T} \delta_t + 2\beta_T \sum_{i=1}^{T} \sigma_{t-1}(x_t) = \mathcal{O}(\sqrt{T}) + \beta_T \mathcal{O}(\sqrt{T\gamma_T}). \quad \text{(by Lemma 3)}$$

## B.3 PROOF OF THEOREMS 3 AND 4

We first introduce a lemma that includes some algebraic derivations, which will be useful for proving the subsequent results.

**Lemma 6** (Appendix C in Chowdhury & Gopalan (2017)). *For any vector $\epsilon$ and let $\hat{\lambda}_t = 1 + 2/t$, the following holds algebraically*

$$\left| k_t(x)^\top (K_{t-1} + \hat{\lambda}_t I)^{-1} \epsilon \right| \leq \hat{\lambda}_t^{-1/2} \sigma_{t-1}(x) \sqrt{\epsilon^\top K_{t-1}(K_{t-1} + \hat{\lambda}_t I)^{-1}\epsilon},$$

$$\epsilon^\top K_{t-1}(K_{t-1} + \hat{\lambda}_t I)^{-1}\epsilon \leq \epsilon^\top \left( (K_{t-1} + (1 - \hat{\lambda}_t)I)^{-1} \right) \epsilon,$$

where $K_{t-1}$ denotes the Gram matrix at time $t$, defined identically as in the main paper but indexed with a subscript to emphasize its dependence on the data available up to time $t-1$. Next, we derive the AF via models constructed by $\mathcal{D}_{t-1} \cup \{(x_{\text{LLM},t}, \tilde{f}_{t-1,s}(x_{\text{LLM},t}))\}$, which we denoted those models as $\mathcal{M}_{t,s} \ \forall s \in I_t$.

**Lemma 7.** *(Lemma 1 in Chen et al. (2025)) Assuming $\mathbb{E}_{\mathcal{M}_{t,s}}[\alpha(x, \mathcal{M}_{t,s})]$ exists, and there exists a function $a : \mathbb{R} \to \mathbb{R}$ such that*

$$\alpha(x; F_{t-1}^+) = \mathbb{E}_{g \sim F_{t-1}^+}[a(g(x))],$$

*then*

$$\alpha(x, F_{t-1}^+) = \mathbb{E}_{\mathcal{M}_{t,s}}[\alpha(x, \mathcal{M}_{t,s})].$$

Lemma 7 arrives at the conclusion that the AF under the $\mathcal{CGP}$ can be computed by the expectation of the AF across all models $\mathcal{M}_{t,s}$ for all $s \in I_t$ under certain conditions. Recall from Lemma 1 that for the $\mathcal{GP}$ constructed using $\mathcal{D}_{t-1}$, previously denoted by $\mathcal{F}_{t-1}$, the difference between the posterior mean $\mu_{t-1}(x)$ and the ground truth function $f(x)$ can be bounded with a suitable $\beta_t$. However, this bound does not directly apply to the $\mathcal{CGP}$, as it is constructed using both historical data and imagined data $(x_{\text{LLM},t}, \tilde{f}_{t-1,s}(x_{\text{LLM},t}))$. The following lemma provides a bound on this difference using a newly constructed $\tilde{\beta}_t$.

**Theorem 5.** *(Theorem 3 in the main paper) Under Assumption 1, for any $\delta \in (0,1)$ and $T \in \mathbb{N}$, with probability at least $1 - \frac{\delta}{T}$, any sample index $s \in I_t$, and any $t$, we have:*

$$|\mu_{t-1,s}^+(x) - f(x)| \leq \tilde{\beta}_t \sigma_{t-1}^+(x),$$

*where $\tilde{\beta}_t = 2B + 2R\sqrt{2(\gamma_t + 1 + \ln(4T/\delta))} + \sqrt{2\ln(4S_t T/\delta)}$.*

*Proof.* As $s$ is fixed and we focusing on deriving the difference between $\mu_{t-1,s}^+(x)$ and $f(x)$, we drop the subscript $s$ for simplicity. Let $k_{t-1}^+$ and $K_{t-1}^+$ denote the kernel vector and Gram matrix, respectively, defined as in Section 2.1, except with the input set augmented to include $x_{\text{LLM},t}$; that is, the input consists of the union of the previously observed designs $x_1, \ldots, x_{t-1}$ and the LLM-suggested point $x_{\text{LLM},t}$. Let $\tilde{\delta} = f(x_{\text{LLM},t}) - \tilde{f}_{t-1}(x_{\text{LLM},t})$, one can express the term $|\mu_{t-1}^+(x) - f(x)|$ as

$$
\begin{aligned}
|\mu_{t-1}^+(x) - f(x)| \leq{} & |f(x) - k_{t-1}^+(x)^\top \left(K_{t-1}^+ + \hat{\lambda}_t I\right)^{-1} [f(x_1), ..., f(x_{t-1}), f(x_{\text{LLM},t})]^\top| \\
& + |k_{t-1}^+(x)^\top (K_{t-1}^+ + \hat{\lambda}_t I)^{-1} [\delta_1, ..., \delta_{t-1}, \tilde{\delta}]^\top| \quad\quad \text{(by (4))} \\
\leq{} & \underbrace{|f(x) - k_{t-1}^+(x)^\top \left(K_{t-1}^+ + \hat{\lambda}_t I\right)^{-1} [f(x_1), ..., f(x_{t-1}), f(x_{\text{LLM},t})]^\top|}_{A} \\
& + \underbrace{|k_{t-1}^+(x)^\top (K_{t-1}^+ + \hat{\lambda}_t I)^{-1} [\delta_1, ..., \delta_{t-1}, 0]^\top|}_{B} \\
& + \underbrace{|k_{t-1}^+(x)^\top (K_{t-1}^+ + \hat{\lambda}_t I)^{-1} [0, ..., 0, \tilde{\delta}]^\top|}_{C}. \quad\quad \text{(by triangle inequality)}
\end{aligned}
$$

Note that terms A and B can be bounded by $B + R\sqrt{2(\gamma_t + 1 + \ln(2T/\delta))}$ with probability at least $1 - \frac{\delta}{2T}$ according to (5). Based on Lemma 6, we can further bound the term $C$ as

$$\left| k_{t-1}^+(x)^\top (K_{t-1}^+ + \hat{\lambda}_t I)^{-1} [0, ..., 0, \tilde{\delta}]^\top \right| \leq \hat{\lambda}_t^{-1/2} \sigma_{t-1}^+(x) \sqrt{[0 \quad \tilde{\delta}] K_{t-1}^+ (K_{t-1}^+ + \hat{\lambda}_t I)^{-1} [0 \quad \tilde{\delta}]^\top}.$$

With probability $1 - \frac{\delta}{4T} - \frac{\delta}{4T} = 1 - \frac{\delta}{2T}$ and by Lemma 6, the square root part of the above equation can be further simplified as

$$\sqrt{\begin{bmatrix} 0 & \tilde{\delta} \end{bmatrix} K_{t-1}^+ (K_{t-1}^+ + \hat{\lambda}_t I)^{-1} \begin{bmatrix} 0 & \tilde{\delta} \end{bmatrix}^\top}$$

$$\leq \sqrt{\begin{bmatrix} 0 & \tilde{\delta} \end{bmatrix} K_{t-1}^+ (K_{t-1}^+ + (1 - \hat{\lambda}_t)I^{-1} + I)^{-1} \begin{bmatrix} 0 & \tilde{\delta} \end{bmatrix}^\top}$$

$$\leq \|\tilde{\delta}\|_2$$

$$\leq |f(x_{\text{LLM},t}) - \tilde{f}_{t-1}(x_{\text{LLM},t})|$$

$$\leq |f(x_{\text{LLM},t}) - \mu_{t-1}(x_{\text{LLM},t})| + |\mu_{t-1}(x_{\text{LLM},t}) - \tilde{f}_{t-1}(x_{\text{LLM},t})|$$

$$\leq (B + R\sqrt{2(\gamma_t + 1 + \ln(4T/\delta))})\sigma_{t-1}(x_{\text{LLM},t})$$

$$\qquad + \sqrt{2\ln(4S_t T/\delta)}\sigma_{t-1}(x_{\text{LLM},t}). \qquad \text{(by Chernoff bound)}$$

Note that $\tilde{f}_{t-1}(x_{\text{LLM},t})$ is sampled from a normal distribution ($F_{t-1}$) with mean $\mu_{t-1}(x_{\text{LLM},t})$ and variance $\sigma_{t-1}^2(x_{\text{LLM},t})$. In this case, one can apply the Chernoff Bound to control the difference between all the samples and the mean response of the $\mathcal{GP}$. As a result, term $C$ can be bounded by $(B + R\sqrt{2(\gamma_t + 1 + \ln(4T/\delta))} + \sqrt{2\ln(4S_t T/\delta)})\sigma_{t-1}^+(x)$ with high probability. Finally, by combining with term A, and with probability $1 - \frac{\delta}{2T} - \frac{\delta}{2T} = 1 - \frac{\delta}{T}$, we have

$$|\mu_{t-1}^+(x) - f(x)| \leq (2B + 2R\sqrt{2(\gamma_t + 1 + \ln(4T/\delta))} + \sqrt{2\ln(4S_t T/\delta)})\sigma_{t-1}^+(x)$$

$$= \tilde{\beta}_t \sigma_{t-1}^+(x),$$

where $\tilde{\beta}_t = 2B + 2R\sqrt{2(\gamma_t + 1 + \ln(4T/\delta))} + \sqrt{2\ln(4S_t T/\delta)}$. $\qquad\square$

**Lemma 8.** *For a set of $S \geq 2$ samples $X_1, \ldots, X_S$, if $|X_s| \leq c, \forall s \in [S]$, then the sample variance satisfies:*

$$\varsigma = \frac{1}{S-1} \sum_{s=1}^{S} (X_s - \bar{X})^2 \leq 2c^2.$$

*Proof.* Let $\bar{X}$ be the sample mean as $\bar{X} = \frac{1}{S} \sum_{s=1}^{S} X_s$. This proof follows the definition of sample variance

$$\varsigma = \frac{1}{S-1} \sum_{s=1}^{S} (X_s - \bar{X})^2 = \frac{1}{S-1} \sum_{s=1}^{S} |X_s - \bar{X}|^2 \leq \frac{S}{S-1} c^2 \leq 2c^2.$$

$\qquad\square$

Now we are ready to derive the upper bound for the cumulative regret. Note that $x_t$ is selected as the maximizer of the $\mathcal{CGP}$-UCB, which means

$$\bar{\mu}_{t-1}(x_t) + \tilde{\beta}_t \sqrt{\sigma_{t-1}^+(x_t)^2 + s_{t-1}^2(x_t)} \geq \bar{\mu}_{t-1}(x) + \tilde{\beta}_t \sqrt{\sigma_{t-1}^+(x)^2 + s_{t-1}^2(x)} \ \forall x \in \mathcal{X}.$$

We first deal with the error cause by $s_{t-1}^2(x)$, which is the sample variance of the predicted mean at $x$, or namely, $k_{t-1}^+(x)(K_{t-1}^+ - \hat{\lambda}_t I)^{-1}(y_1, ..., y_{t-1}, \tilde{f}_{t-1,s}(x_{\text{LLM},t}))^\top \ \forall s \in I_t$. Note that there is no uncertainty in $k_{t-1}^+(x)(K_{t-1}^+ - \hat{\lambda}_t I)^{-1}$ and also $(y_1, ..., y_{t-1})$, hence we can substract it and simply consider the variance of

$$k_{t-1}^+(x)(K_{t-1}^+ - \hat{\lambda}_t I)^{-1} \begin{bmatrix} 0 & \tilde{f}_{t-1,s}(x_{\text{LLM},t}) \end{bmatrix}^\top \ \forall s \in I_t.$$

In order to apply Lemma 8, we first derive the upper bound for $k_{t-1}^+(x)(K_{t-1}^+ - \hat{\lambda}_t I)^{-1} \begin{bmatrix} 0 & \tilde{f}_{t-1,s}(x_{\text{LLM},t}) - M \end{bmatrix}^\top \ \forall s \in I_t$, where $M = \frac{1}{|I_t|} \sum_{s \in I_t} \tilde{f}_{t-1,s}(x_{\text{LLM},t})$. With

probability$1 - \frac{\delta}{4T}$ and by Lemma 6, we have

$$k_{t-1}^+(x)(K_{t-1}^+ - \hat{\lambda}_t I)^{-1} \begin{bmatrix} 0 & \tilde{f}_{t-1,s}(x_{\text{LLM},t}) - M \end{bmatrix}^\top$$

$$\leq \hat{\lambda}_t^{-1/2} \sigma_{t-1}^+(x) \sqrt{[0, \tilde{f}_{t-1,s}(x_{\text{LLM},t}) - M]^\top (K_{t-1}^+ + \hat{\lambda}_t I)^{-1} [0, \tilde{f}_{t-1,s}(x_{\text{LLM},t}) - M]}$$

$$\leq \hat{\lambda}_t^{-1/2} \sigma_{t-1}^+(x) \sqrt{(\tilde{f}_{t-1,s}(x_{\text{LLM},t}) - M)^2}$$

$$\leq \hat{\lambda}_t^{-1/2} \sigma_{t-1}^+(x) \sqrt{(\tilde{f}_{t-1,s}(x_{\text{LLM},t}) - \mu_{t-1}(x_{\text{LLM},t}))^2}$$

$$= \hat{\lambda}_t^{-1/2} \sigma_{t-1}^+(x) |\tilde{f}_{t-1,s}(x_{\text{LLM},t}) - \mu_{t-1}(x_{\text{LLM},t})|$$

$$\leq \sigma_{t-1}^+(x) \sqrt{2 \ln(4 S_t T / \delta)},$$

where the last inequality uses the fact that $\hat{\lambda} \leq 1$ and by the Chernoff Bound. In this case, by Lemma 8, the variance of $k_{t-1}^+(x)(K_{t-1}^+ - \hat{\lambda}_t I)^{-1} \begin{bmatrix} 0 & \tilde{f}_{t-1,s}(x_{\text{LLM},t}) \end{bmatrix}^\top \; \forall s \in I_t$ can be bounded as

$$s_{t-1}^2(x) \leq 4\sigma_{t-1}^+(x)^2 \ln(4 S_t T / \delta). \tag{7}$$

Note that by Theorem 5, the ground truth $f(x_t)$ can be bounded by $\mu_{t-1,s}^+(x) \pm \tilde{\beta}_t \sigma_{t-1}^+(x)$ with high probability for all index $s$ in $I_t$, this also holds for the mean over all $s \in I_t$, that is,

$$\bar{\mu}_{t-1}^+(x) - \tilde{\beta}_t \sigma_{t-1}^+(x) \leq f(x) \leq \bar{\mu}_{t-1}^+(x) + \tilde{\beta}_t \sigma_{t-1}^+(x).$$

With probability at least $1 - \delta$, we can derive the upper bound for $r_t = f(x^*) - f(x_t)$ as

$$\begin{aligned}
r_t &= f(x^*) - f(x_t) \\
&\leq \bar{\mu}_{t-1}^+(x^*) + \tilde{\beta}_t \sigma_{t-1}^+(x^*) - \left( \bar{\mu}_{t-1}^+(x_t) - \tilde{\beta}_t \sigma_{t-1}^+(x_t) \right) \\
&= \left( \bar{\mu}_{t-1}^+(x^*) - \bar{\mu}_{t-1}^+(x_t) \right) + \tilde{\beta}_t \sigma_{t-1}^+(x^*) + \tilde{\beta}_t \sigma_{t-1}^+(x_t) \\
&\leq \tilde{\beta}_t \sqrt{\sigma_{t-1}^+(x_t)^2 + s_{t-1}^2(x_t)} - \tilde{\beta}_t \sqrt{\sigma_{t-1}^+(x^*)^2 + s_{t-1}^2(x^*)} + \tilde{\beta}_t \sigma_{t-1}^+(x^*) + \tilde{\beta}_t \sigma_{t-1}^+(x_t) \\
&\leq \tilde{\beta}_t \sigma_{t-1}^+(x_t) + \tilde{\beta}_t s_{t-1}(x_t) - \tilde{\beta}_t \sigma_{t-1}^+(x^*) + \tilde{\beta}_t \sigma_{t-1}^+(x^*) + \tilde{\beta}_t \sigma_{t-1}^+(x_t) \\
&= 2\tilde{\beta}_t \sigma_{t-1}^+(x_t) + \tilde{\beta}_t s_{t-1}(x_t) \\
&\leq \mathcal{O}(\sqrt{\gamma_t + \ln(t)}) \sigma_{t-1}^+(x_t) + \mathcal{O}(\sqrt{\gamma_t} \ln(t)/t) \sigma_{t-1}^+(x_t) \qquad \text{(by (7) and Theorem 5)} \\
&\leq \mathcal{O}(\sqrt{\gamma_t + \ln(t)} \sigma_{t-1}^+(x_t).
\end{aligned}$$

The cumulative regret can be bounded as

$$\begin{aligned}
R_t = \sum_{i=1}^T r_t &= \sum_{i=1}^T \mathcal{O}(\sqrt{\gamma_t + \ln(t)}) \sigma_{t-1}^+(x_t) \\
&\leq \mathcal{O}(\sqrt{\gamma_T + \ln(T)}) \sum_{i=1}^T \sigma_{t-1}^+(x_t) \\
&\leq \mathcal{O}(\sqrt{\gamma_T + \ln(T)}) \mathcal{O}(\sqrt{T \gamma_T}) \qquad \text{(by Lemma 3)} \\
&= \mathcal{O}(\sqrt{T \gamma_T (\gamma_T + \ln(T))}).
\end{aligned}$$

## C  ADDITIONAL EXPERIMENT

### C.1  COMPARISON WITH HAIC BO

While the scale of external information considered in previous HAIC works is not directly comparable to the setting of either *LLM-assisted BO* or the proposed methods, in this section we compare our approach with $\pi$BO (Hvarfner et al. (2022)) on the Branin-$2D$ and Levy-$2D$ functions by replace human's effort on suggesting $\pi(x)$, the preference function, using LLMs. Specifically, we provide the problem context and the initial dataset as input to the LLM. For each function (defined on $[0, 1]^2$),

we then randomly select 100 points $\{z_i\}_{i=1}^{100}$ and query the LLM for the probability of each point being the optimum, denoted $p_i$, $i \in [100]$. To approximate a continuous prior $\pi(x)$, we normalize the probabilities to sum to one and apply Kernel Density Estimation. The hyperparameter $\beta$ is set to $T/100$, following the settings in Hvarfner et al. (2022), and all other configurations remain the same as in Section 3. The acquisition function in $\pi$BO is given by

$$\alpha_\pi(x, F_{t-1}) = \alpha(x, F_{t-1})\, \pi(x)^{\beta/t},$$

where $\alpha$ is the acquisition function, which we set to UCB in this experiment. We also evaluate a dynamic variant in which $\pi(x)$ is updated at each iteration by re-querying the LLM with both the problem context and the historical observations, where we call it $\pi$BO-dynamic.

Figure 6 presents the regret trajectories for all methods. We observe that our proposed approaches consistently outperform $\pi$BO in both experimental settings. Notably, even though $\pi$BO updates the preference function $\pi(x)$ at every iteration using the LLM, its performance remains unstable and unreliable. We acknowledge that extracting richer information from LLMs—beyond a single design point per iteration—remains an open question and represents an exciting direction for future research.

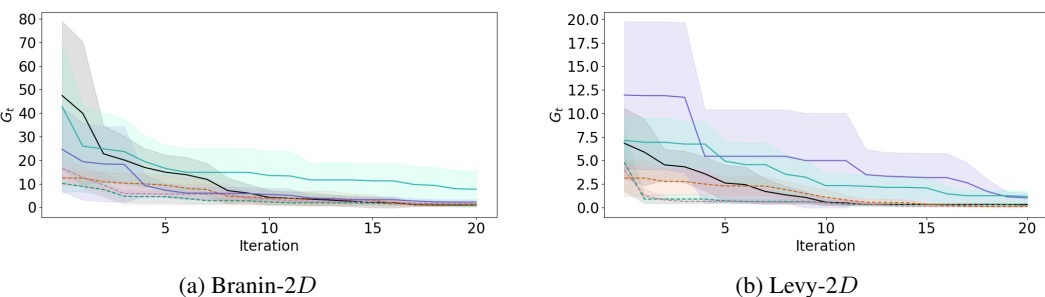

(a) Branin-2$D$          (b) Levy-2$D$

Figure 6: Regret comparison between proposed methods and $\pi$BO. Each line shows the regret $G_t$, shaded with 95% confidence intervals. **Proposed methods:**`---` `LLINBO-Transient`, `---` `LLINBO-Justify`, `---` `LLINBO-Constrained`. **Baselines:** —— $\pi$BO, —— $\pi$BO-dynamic —— BO.

## C.2 EXPERIMENTS ON HIGH DIMENSIONAL SETTINGS

In this section, we evaluate the proposed methods on two BBO tasks using the Levy-15$D$ and Ackley-12$D$ benchmark functions. All experimental settings—including hyperparameters, number of replications, LLM agents, GP configurations, and the size of the initial design—are kept identical to those described in Section 3, except for the budget, which is set to $T = 100$.

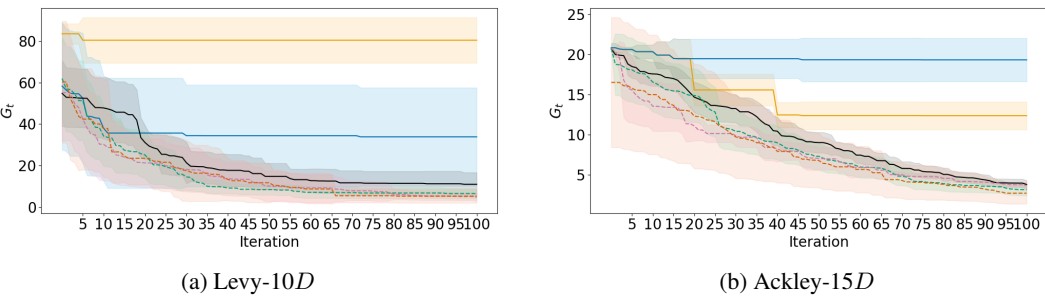

(a) Levy-10$D$          (b) Ackley-15$D$

Figure 7: $G_t$ comparison for BBO. Each line shows the mean regret, shaded with 95% confidence intervals. **Proposed methods:** `---` `LLINBO-Transient`, `---` `LLINBO-Justify`, `---` `LLINBO-Constrained`. **Baselines:** —— LLAMBO, —— LLAMBO-light, —— BO.

The plots in Fig.7 compare the regret curves over 100 iterations on two benchmark functions, Levy-10$D$ and Ackley-15$D$. The proposed `LLINBO` variants consistently decrease regret faster and more steadily than the baselines, showing both lower mean regret and tighter confidence intervals. In contrast, the LLAMBO-based baselines remain much higher and flatter, indicating slower improvement

and greater uncertainty throughout the optimization process. We hypothesize that this behavior arises from the increased prompt length in higher-dimensional problems, which reduces the LLM's ability to consistently concentrate on the optimal region.

### C.3 EXPERIMENTS ON THE DYNAMICS OF `LLINBO-JUSTIFY`

The key to the trustworthiness of `LLINBO-Justify` lies in its ability to leverage LLMs only when their recommendations are deemed valuable, while discarding them whenever the statistical surrogate model strongly believes that such suggestions would lead to inferior performance. In this experiment, we would like to access this property on two different scenarios: BBO task using Levy-$2D$ and HPT task using Piston with XGB-$4D$.

From Figure 3, we observe that LLAMBO-light, the LLM agent embedded in `LLINBO-Justify`, performs well on the Levy-$2D$ function, in contrast to its behavior on the Piston with XGB-$4D$ task in Figure 4, where the regret remains almost constant after approximately six iterations. In this experiment, we fix all parameters, LLM agent, BO settings, and initial data size, to be identical to those in Section 3. Our goal is to quantify how frequently `LLINBO-Justify` accepts LLM-generated suggestions. For clearer visualization, we set the optimization horizon to $T = 20$ for both Levy-$2D$ and Piston with XGB-$4D$.

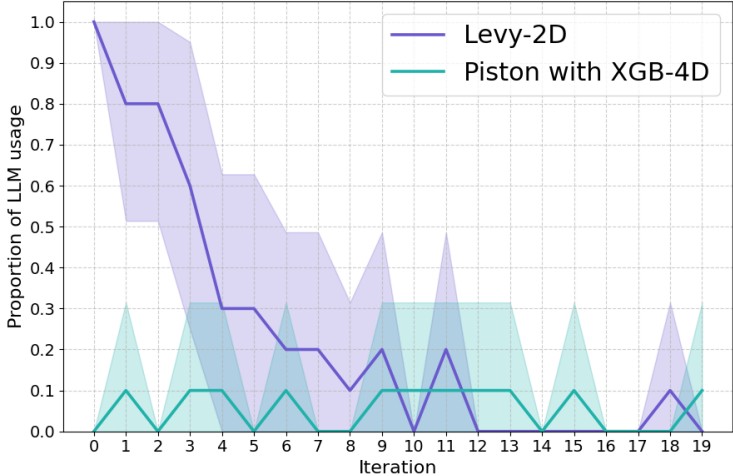

Figure 8: Proportion when LLM is used in `LLINBO-Justify` between two tasks. Each line shows the proportion of the 10 repeated experiments where LLM's suggestion is used as the next design at a specific iteration, shaded with 95% confidence intervals. **Tasks**: —— Levy-$2D$, —— Piston with XGB-$4D$.

We repeat each setting 10 times and record, for every iteration and every repetition, whether the LLM's suggestion is selected as the next design point. Figure XX reports, for each iteration, the proportion of runs in which the LLM suggestion was used. The results show that `LLINBO-Justify` relies on LLM suggestions much more frequently in the Levy-$2D$ case, but far less in the Piston with XGB-$4D$ setting. This behavior aligns with our expectations: the algorithm allows the LLM to guide the optimization when its proposals appear promising, while discarding them otherwise.

Finally, aggregating across all repeated experiments, the proportion of iterations in which the LLM's suggestion was used is $0.295 \pm 0.0548$ for Levy-$2D$, but only $0.105 \pm 0.222$ for Piston with XGB-$4D$.

### C.4 INFLUENCE OF PROMPT INFORMATIVENESS ON `LLINBO`

In this section, we investigate how the amount of contextual information provided to the LLM agent affects the performance of our proposed `LLINBO` variants. Although the main focus of this work is on integrating LLM agents into the BO framework, we appreciate the reviewer's suggestion to explore prompt ablations. This section presents additional experiments designed to assess the sensitivity of our methods to different levels of LLM informativeness.

To analyze the influence of LLM knowledge, we conducted HPT task on Robot with RF-4$D$. We modified the prompt content used in **LLAMBO-Light**, which serves as the LLM agent in our framework, under the following three settings:

- **Fully informed:** The prompt includes historical data, extracted data patterns, and Random Forest model patterns.
- **Partially informed:** The prompt includes historical data and model patterns.
- **Minimally informed:** The prompt includes only the historical dataset.

All other experimental configurations were kept identical to those in the main paper. Performance is summarized using the proportion of improvement,

$$I_t = \frac{\text{regret at } t = 0 - \text{regret at } t = T}{\text{regret at } t = 0},$$

where higher values indicate better optimization performance. For each setting, we conducted ten independent repetitions and report the mean and standard deviation in Table 2.

| Method | Fully informed | Partially informed | Minimally informed |
|---|---|---|---|
| LLINBO-Transient | $0.055 \pm 0.011$ | $0.052 \pm 0.011$ | $0.040 \pm 0.012$ |
| LLINBO-Justify | $0.065 \pm 0.031$ | $0.059 \pm 0.029$ | $0.062 \pm 0.029$ |
| LLINBO-Constrained | $0.061 \pm 0.021$ | $0.060 \pm 0.017$ | $0.062 \pm 0.022$ |

Table 2: Performance comparison under different levels of LLM contextual information. Values are means and standard deviations over ten repetitions.

The performance of **LLINBO-Transient** is noticeably affected by the informativeness of the LLM inputs. When the LLM receives limited contextual information, the optimization performance declines. However, as the design process gradually shifts from the LLM to the GP through the diminishing schedule of $p_t$, the final regret converges across all settings. This behavior confirms the importance of the diminishing-$p_t$ design, which reduces the algorithm's dependence on potentially noisy LLM guidance and enhances robustness.

In contrast, **LLINBO-Justify** consistently safeguards against poor LLLM suggestions through its client-level, data-driven acceptance–rejection mechanism. Interestingly, its performance improves in the minimally informed setting, highlighting the strength of validating each LLM suggestion using surrogate uncertainty. This mechanism effectively mitigates risks caused by unreliable or noisy LLM recommendations.

Finally, **LLINBO-Constrained** demonstrates strong robustness. When the LLM provides limited or unhelpful information, the algorithm automatically defaults to classical BO steps, preventing any deterioration in performance.

## D   SELECTION BETWEEN THE PROPOSED ALGORITHMS AND HYPERPARAMETERS

**Choosing between the proposed methods.**   It is noteworthy that the regret bounds for all three methods contain no variables or assumptions on the LLMs, thereby ensuring the **no-harm guarantees** introduced by Xu et al. (2024). In other words, the quality of LLM suggestions does not degrade their performance, and the choice among them can therefore be guided by practical needs. LLINBO-Transient is the most interpretable and practical for non-expert users, employing an explicit temporal schedule to reduce LLM influence over time. Importantly, the reliance on LLM suggestions diminishes as the probability of querying the LLM approaches zero, making this variant suitable for practitioners prioritizing transparency, simplicity, or scenarios where accessing LLMs is costly. LLINBO-Justify adopts a more data-driven approach by learning a justification threshold for LLM suggestions without altering the BO machinery, thereby maintaining interpretability while offering adaptive control—an attractive option when flexibility is desired without structural changes. Finally, LLINBO-Constrained is the most robust and theoretically grounded variant, integrating LLMs and BO through a probabilistic constraint that automatically hedges against finite-sample

uncertainty and requires no additional hyperparameter tuning, making it particularly well-suited for safety-critical or resource-constrained settings where minimizing risk and avoiding hyperparameter tuning are essential.

**Selecting hyperparameters.** We acknowledge that leveraging LLMs for BO is still in its early stages. As such, tuning the algorithm's parameters based on the LLM's level of understanding remains an open but important research direction. Nevertheless, we outline below general-purpose strategies for selecting these parameters. For $p_t$ in `LLINBO-Transient`, our approach introduces a diminishing reliance on the LLM over time. We therefore set $1 - p_t \in \mathcal{O}(1/t^2)$, which limits the influence of potentially unreliable LLM suggestions as optimization progresses. Indeed, we consistently observed that the LLM's ability to exploit diminishes rapidly over time, unsurprising since LLMs lack explicit surrogate modeling and calibrated uncertainty; however, when the problem domain is well understood by the LLM (e.g., hyperparameter tuning on standard datasets), the increase in $p_t$ can be made more gradual. In contrast, the performance of the LLM is less critical in `LLINBO-Justify`, as this variant is primarily data-driven and can automatically hedge against unreliable suggestions. Following our theoretical results, we recommend using a conservative decreasing schedule for $\psi_t$ ($\mathcal{O}(1/t)$, Theorem 2 in the main paper) and setting $\psi_0$ in a way that reflects the structure of the acquisition function. For instance, when using UCB, $\psi_0$ can be the posterior variance at the first LLM-suggested point, or in the case of Thompson Sampling, the difference between the maximum and minimum values in a posterior sample. Finally, `LLINBO-Constrained` was specifically designed to minimize the need for hyperparameter tuning, with the only parameter being the sampling size from the constrained GP, which should be dictated by available computational resources. We recommend starting with the largest feasible sample size and then gradually reducing it (as permitted by our theory) based on constraints. In our implementation, we began with a large sample size and reduced it at a rate of $\mathcal{O}(1/t^2)$, which offered a good balance between computational efficiency and performance.

## E   COMPUTATIONAL COMPLEXITY OF LLINBO

In this section, we analyze the computational complexity of the three `LLINBO` variants—both mathematically and empirically—and discuss the tradeoff between computational efficiency and optimization performance, with particular emphasis on `LLINBO-Constrained`.

Let $C_{\mathrm{LLM}}$ denote the computational cost of querying the LLM, and let $C_{\mathcal{GP}}$ represent the cost associated with extracting the next design point from the GP surrogate. For simplicity, we assume that all LLM-related operations—including warm-starting and candidate sampling—incur the same cost $C_{\mathrm{LLM}}$. Likewise, we assume all $\mathcal{GP}$-related operations—sampling candidate points and optimizing the acquisition function—incur a uniform cost $C_{\mathcal{GP}}$.

**Handover property of `LLINBO-Transient`.** By construction, the expected computational cost of `LLINBO-Transient` up to time $T$ is

$$\mathbb{E}[C_{\mathrm{LLINBO\text{-}Transient}}] = (T_0 + \sum_{t=1}^{T}(1 - p_t)) C_{\mathrm{LLM}} + \sum_{t=1}^{T} p_t\, C_{\mathcal{GP}} = \mathcal{O}(\log T)C_{\mathrm{LLM}} + \mathcal{O}(T)C_{\mathcal{GP}}.$$

Thus, the LLM-related computation grows only sublinearly, which is desirable in modern BO pipelines where LLM inference is typically more expensive—in both time and monetary cost—than GP-based inference. The GP-related cost naturally scales as $\mathcal{O}(T)$, matching the behavior of classical BO.

We further formalize the frequency with which `LLINBO-Transient` queries the LLM in the following lemma.

**Lemma 9.** *Let $Q_t \in \{0, 1\}$ denote whether the algorithm queries the LLM at iteration $t$, with*

$$\mathbb{P}(Q_t = 1 \mid \mathcal{F}_{t-1}) = 1 - p_t,$$

*and assume $1 - p_t = \mathcal{O}(1/t)$. Let*

$$I_T = \sum_{t=1}^{T} Q_t$$

*denote the cumulative number of LLM queries up to iteration $T$. Then, with probability at least $1 - \delta$,*

$$I_T = \mathcal{O}(\sqrt{T}) \quad as \; T \to \infty.$$

*Thus, LLINBO-Transient hands over to standard BO with high probability.*

*Proof.* Define the martingale

$$Y_t = \sum_{s=1}^{t} \Big( Q_s - (1 - p_s) \Big), \qquad Y_0 = 0.$$

Then $\mathbb{E}[Y_t \mid \mathcal{F}_{t-1}] = Y_{t-1}$, and the increments satisfy $|Y_t - Y_{t-1}| = |Q_t - (1 - p_t)| \leq 1$ almost surely. Applying the Azuma–Hoeffding inequality (Lemma 5), we obtain

$$\Pr\big(Y_T \geq \epsilon\big) \leq \exp\Big( -\frac{\epsilon^2}{2T} \Big).$$

Thus, with probability at least $1 - \delta$,

$$Y_T \leq \sqrt{2T \log(1/\delta)}.$$

Since $I_T = \sum_{t=1}^{T}(1 - p_t) + Y_T$ and $\sum_{t=1}^{T}(1 - p_t) = \mathcal{O}(\log T)$, we conclude that

$$I_T \leq \mathcal{O}(\log T) + \sqrt{2T \log(1/\delta)} = \mathcal{O}(\sqrt{T})$$

with probability at least $1 - \delta$. Therefore, the cumulative number of LLM calls is sublinear, implying that LLINBO-Transient eventually relies primarily on GP-based BO. $\square$

**Computational Complexity of LLINBO–Justify.** Unlike LLINBO-Transient, the LLINBO-Justify variant requires querying *both* the LLM and the GP at every iteration. Thus, its expected computational cost is

$$\mathbb{E}[C_{\texttt{LLINBO-Justify}}] = (T_0 + T) \, C_{\text{LLM}} + T \, C_{\mathcal{GP}}.$$

Consequently, LLINBO-Justify incurs a higher cost than classical BO, LLAMBO-light (the LLM agent alone), and LLINBO-Transient.

**Computational Complexity of LLINBO–Constrained.** The LLINBO-Constrained method additionally requires generating multiple GP-based samples per iteration to enforce safety constraints. Its expected complexity is

$$\mathbb{E}[C_{\texttt{LLINBO-Constrained}}] = (T_0 + T) \, C_{\text{LLM}} + \sum_{t=1}^{T} S_t \, C_{\mathcal{GP}} = \mathcal{O}(T)C_{\text{LLM}} + \mathcal{O}(T + \log T)C_{\mathcal{GP}},$$

where $S_t$ denotes the number of surrogate evaluations at iteration $t$. Because $S_t$ typically grows with the number of clients or safety evaluations, this variant is the most computationally demanding among the three.

Assuming $C_{\text{LLM}} > C_{\mathcal{GP}}$, which reflects the common cost hierarchy in practice, we obtain the following ordering of computational complexity:

$$C_{\texttt{LLINBO-Constrained}} > C_{\texttt{LLINBO-Justify}} > C_{\text{LLAMBO-Light}} > C_{\texttt{LLINBO-Transient}} > C_{\text{BO}}.$$

This ordering highlights a fundamental tradeoff between computational cost and performance improvements through LLM-guided exploration. While LLINBO-Constrained is the most computationally intensive, it provides robustness guarantees absent in the lighter methods. Conversely, LLINBO-Transient offers strong practical efficiency while still benefiting from occasional LLM guidance.

| Method | Experiment | Time (s) |
|---|---|---|
| BO | Rastrigin-2D | $12.19 \pm 1.12$ |
| | Robot with RF-4D | $48.76 \pm 3.82$ |
| LLAMBO | Rastrigin-2D | $907 \pm 12$ |
| | Robot with RF-4D | $3628 \pm 48$ |
| LLAMBO-light | Rastrigin-2D | $144 \pm 5$ |
| | Robot with RF-4D | $432 \pm 11$ |
| LLINBO-Transient | Rastrigin-2D | $92.59 \pm 11.12$ |
| | Robot with RF-4D | $278 \pm 18.12$ |
| LLINBO-Justify | Rastrigin-2D | $167 \pm 4$ |
| | Robot with RF-4D | $668 \pm 8$ |
| LLINBO-Constrained | Rastrigin-2D | $224 \pm 25.63$ |
| | Robot with RF-4D | $896 \pm 31$ |

Table 3: Summary of computational time (in seconds) across methods for two benchmark experiments, averaged over 10 runs.

**Empirical computation time.** We evaluate the computational overhead of each method using two benchmark tasks: Rastrigin-$2D$ for Bayesian black-box optimization (BBO) and Robot with RF-$4D$ for hyperparameter tuning (HPT). All settings follow those used in the main experiments, and each experiment is repeated 10 times. The reported runtimes correspond to the wall-clock time recorded separately for each run. Experiments were conducted on a system with 5 nodes, each equipped with dual Intel Xeon Platinum CPUs and 512 GB of RAM. The results are summarized in Table 3.

Based on Table 3, several observations can be made. First, the empirical results are consistent with our theoretical analysis of computational complexity: LLINBO-Constrained is the most computationally expensive among the LLINBO variants, while LLINBO-Transient is the most efficient. LLAMBO exhibits the highest runtime overall, due to repeated interactions with both the LLM and surrogate model at each iteration. Even though LLAMBO was implemented with parallelism, it remains significantly slower than other methods. In contrast, LLAMBO-light is substantially more efficient, as it avoids the repeated GP updates needed in full LLAMBO.

We also observe higher variance in computational time for the LLM-based methods compared to BO. This can be attributed to occasional failures in LLM responses, such as format errors or mismatched dimensions, which require re-querying. Furthermore, the large standard deviation for LLINBO-Constrained is expected: when the number of retained samples is large, evaluating the aggregated posterior mean and variance (as described in Theorem 3) dominates the computation for that iteration, leading to increased variability across runs.

**Trade-off between complexity and performance in LLINBO-Constrained.** An experiment on BBO task using Rastrigin-$2D$ is performed to assess the trade-off between computational complexity and the regret in LLINBO-Constrained. More specifically, we aim to link the settings of $S_1$ (initial sample size) and the decreasing rate of $S_t$ to the performance. We consider $S_1 = 100, 1000, 5000$ and $S_t = S_1, S_t = S_1/t, S_t = S_1/t^2$. All other settings are the same as in the main paper. Fig. 9 shows the regret curves for each setting compared with the two baselines: BO and LLAMBO-light.

We can derive several important insights from Fig. 9. First, BO consistently reduces regret across all iterations, while the LLM-based agent (LLAMBO-light) is highly effective in the early phase but struggles to provide meaningful improvements thereafter. This highlights a fundamental limitation of LLM-guided exploration—strong initial performance followed by diminishing returns.

Second, the behavior of LLINBO-Constrained becomes increasingly similar to standard BO as the decay rate of $S_t$ increases. When $S_t$ is constant, the algorithm remains partially influenced by the LLM's suggestions, which can be suboptimal in later stages. However, setting $S_t = S_1/t^2$ yields regret curves that closely align with BO, indicating that a faster decay reduces reliance on the LLM at later iterations, when its suggestions become less reliable. This finding supports the use of an aggressive decay schedule for $S_t$.

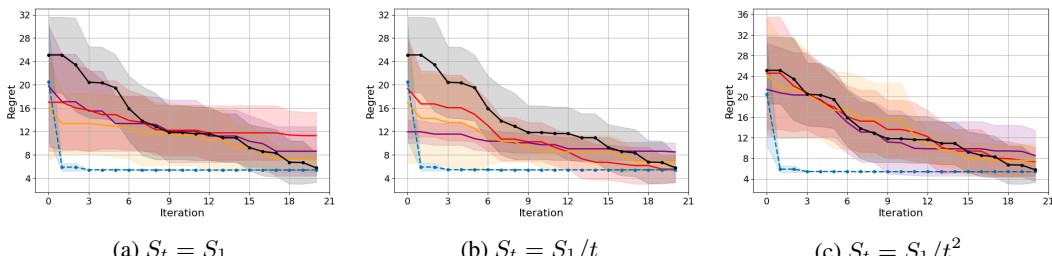

(a) $S_t = S_1$        (b) $S_t = S_1/t$        (c) $S_t = S_1/t^2$

Figure 9: $G_t$ comparison on Rastrigin-2$D$ using different settings of $S_1$ and $S_t$. Each line shows the mean regret, shaded with 95% confidence intervals. **Proposed methods:** —— LLINBO-Constrained with $S_1 = 100$, —— LLINBO-Constrained with $S_1 = 1000$, —— LLINBO-Constrained with $S_1 = 5000$. **Baselines:** —— BO, —— LLAMBO-light.

Third, varying $S_1$ reveals a useful trade-off between early-stage performance and long-term robustness. A larger initial $S_1$ enables the algorithm to leverage the LLM's few-shot learning strengths, as seen in the yellow and purple curves in Fig. 9(a) and Fig. 9(b). These observations are consistent with our design choices in Section 3, where we adopt $S_t = 10000/t^2$, and align with the practical guidelines provided in Appendix D. In practice, we recommend setting $S_1$ based on available computational resources and adopting a decay rate on the order of $\mathcal{O}(1/t^2)$.

Finally, while the computational–performance trade-off is difficult to quantify precisely due to the black-box nature of LLMs, the trend in the regret curves offers intuitive guidance. Since LLMs perform well in the early iterations but fail to exploit in the later phase, setting a large $S_1$ (to fully utilize initial LLM strength) and decreasing it over time (to prioritize exploitation and reduce computational cost) provides a balanced and practical strategy for LLINBO-Constrained.

# F NUMERICAL EXPERIMENTS DETAILS

We utilize GPT-3.5-turbo as the LLM agent, selected for its demonstrated capability to generate high-quality responses. The temperature parameter is set to its default value of 1.0. Prompt structures for LLAMBO are primarily adapted from the methodology proposed by Liu et al. (2024). For each task, we define a task-specific system prompt. Specifically, the system prompt for BBO is: *"You are an AI assistant that helps people find the maximum of a black-box function."* and for hyperparameter tuning tasks: *"You are an AI assistant that helps me reduce the mean square error by tuning the hyperparameters in a machine learning model."*

We use SingleTaskGP in Python's BOTorch package Balandat et al. (2020) as the surrogate model when a statistical model is involved. Namely, its prior mean is set to be constant, where the constant is learned while training, and the kernel function is set to be matern 5/2 with automatic relevance determination.

## F.1 EXPERIMENTAL DETAILS FOR BBO

For the BBO task, we employ the following simulation functions: Levy-2$D$, Rastrigin-2$D$, Branin-2$D$, Bukin-2$D$, Hartmann-4$D$, and Ackley-6$D$, as implemented in the Virtual Library of Simulation Experiments Surjanovic & Bingham (2013). Each function is rescaled to the unit hypercube $[0, 1]^D$, and a negative sign is applied to the response to convert the problem into a maximization task. A summary of these simulation functions is provided below.

- Levy-2$D$

$$w_i = 1 + \frac{x_i - 0.5}{4}, \quad i = 1, 2$$

$$f(x) = -\sin^2(\pi w_1) - \sum_{i=1}^{1}(w_i - 1)^2\left[1 + 10\sin^2(\pi w_i + 1)\right] - (w_2 - 1)^2\left[1 + \sin^2(2\pi w_2)\right]$$

- Rastrigin-$2D$

$$x' = 10.24x - 5$$

$$f(x) = -12 - \sum_{i=1}^{2} \left[ x'^2_i - 10\cos(2\pi x'_i) \right]$$

- Branin-$2D$

$$x'_1 = 15x_1 - 5, \quad x'_2 = 15x_2$$

$$f(x) = -\left( x'_2 - \frac{5.1}{4\pi^2}x'^2_1 + \frac{5}{\pi}x'_1 - 6 \right)^2 - 10\left( 1 - \frac{1}{8\pi} \right)\cos(x'_1) - 10$$

- Bukin-$2D$

$$x'_1 = 20x_1 - 15, \quad x'_2 = 6x_2 - 3$$

$$f(x) = -100\sqrt{|x'_2 - 0.01x'^2_1|} - 0.01\,|x'_1 + 10|$$

- Hartmann-$4D$

$$f(x) = -\sum_{i=1}^{4} a_i \exp\left( -\sum_{j=1}^{4} A_{ij}(x_j - P_{ij})^2 \right)$$

With constants:

$$a = [1.0, 1.2, 3.0, 3.2]$$

$$A = \begin{bmatrix} 10 & 3 & 17 & 3.5 \\ 0.05 & 10 & 17 & 0.1 \\ 3 & 3.5 & 1.7 & 10 \\ 17 & 8 & 0.05 & 10 \end{bmatrix}$$

$$P = 10^{-4} \times \begin{bmatrix} 1312 & 1696 & 5569 & 124 \\ 2329 & 4135 & 8307 & 3736 \\ 2348 & 1451 & 3522 & 2883 \\ 4047 & 8828 & 8732 & 5743 \end{bmatrix}$$

- Ackley-$6D$

$$f(x) = -20\exp\left( -0.2\sqrt{\frac{1}{6}\sum_{i=1}^{6}x_i^2} \right) - \exp\left( \frac{1}{6}\sum_{i=1}^{6}\cos(2\pi x_i) \right) + 20 + e$$

**Prompts design for BBO task.** To facilitate effective reasoning by the LLM, each function is accompanied by a Description Card , which provides essential contextual information. The Description Card includes the following components:

- `Function Patterns`: A high-level summary of the function's characteristics, offering partial information to guide the LLM's reasoning. For example:

  *"Non-convex and multi-modal. The function exhibits a nearly flat outer region with a prominent central depression, resulting in multiple local optima surrounding a single global optimum. It is highly symmetric and separable, yet optimization remains challenging due to the abundance of local maxima."*

- `Dimensionality`: Specifies the number of input dimensions. Given that the input space is normalized to the unit hypercube, this field simply indicates the dimensionality of the design space.

The `Function Patterns` included in each Description Card are derived from the benchmark function descriptions provided by Surjanovic & Bingham (2013), and a summary of these patterns is presented in Table 4.

Next, we introduce Data Card , which collects the information of previously observed designs and the responses. For example, at iteration 4, the Data Card would be *x: (0.2334, 0.12), f(x): 1.2311; x: (0.1217, 0.433), f(x): 1.091; x: (0.9, 0.5), f(x): 4.502; x: (0.108, 0.203), f(x): 3.22.*

| Simulation functions | Description Card [Function Patterns] |
|---|---|
| Levy-$2D$ | highly multimodal but with a unique global maximum. |
| Rastrigin-$2D$ | which is highly multimodal, non-convex function with a large number of regularly spaced local minima. |
| Branin-$2D$ | smooth, multimodal benchmark with three global maxima |
| Bukin-$2D$ | steep, narrow, and highly non-convex landscape with a sharp valley and a unique global maximum |
| Hartmann-$4D$ | 4-dimensional, non-convex, multi-modal and is composed of weighted, anisotropic Gaussian-like bumps centered at different points, making it highly non-separable and challenging to optimize. |
| Ackley-$6D$ | 6-dimensional, non-convex, and multi-modal. The function exhibits a nearly flat outer region and a large hole at the center, resulting in many local optima surrounding a single global optimum. It is highly symmetric and separable in nature, but optimization is still challenging due to the numerous local maxima. |

Table 4: Function patterns used in the **Description Card** for each simulation function.

In the `LLAMBO` framework, candidate sampling is facilitated by a structured prompt designed to elicit a diverse set of potential query points. This mechanism is illustrated in the Candidate sampling phase of Table 5. At each iteration, we prompt LLM 10 times to generate a total of 10 candidate points. To enhance the diversity of these candidates, we follow the strategy outlined in Liu et al. (2024), where the content of the **Data Card** is permuted across prompts.

The `LLAMBO` framework Liu et al. (2024) introduces a hyperparameter $\alpha = 0.1$ to balance exploration and exploitation during the candidate sampling phase. At iteration $t$, we compute the **Target Score** based on the current observed values $\{y_i\}$ as follows:

$$\textbf{Target Score} = \begin{cases} \min_i y_i + \alpha \cdot (\max_i y_i - \min_i y_i), & \text{for minimization,} \\ \max_i y_i - \alpha \cdot (\max_i y_i - \min_i y_i), & \text{for maximization.} \end{cases}$$

This value serves as a dynamic threshold to guide the LLM in proposing candidates that are both competitive with current best observations and diverse enough to enable exploration.

In the `LLAMBO` framework, a surrogate prompt is used to estimate the predictive mean and variance at each candidate point generated by the candidate sampling prompt. This process corresponds to the Surrogate modeling phase illustrated in Table 5. To promote variability in the surrogate responses, we similarly permute the **Data Card** across prompts. Finally, an AF is applied to select the next query point. We adopt the Expected Improvement (EI) criterion Jones et al. (1998), consistent with the acquisition strategy employed in Liu et al. (2024).

In contrast, the `LLAMBO-light` variant bypasses explicit surrogate querying by prompting LLM directly with the problem formulation and historical observations to generate the next evaluation point. This streamlined design process corresponds to the Candidate generation phase shown in Table 5.

| Phases | Prompts |
|---|---|
| Warmstarting `LLAMBO` `LLAMBO-light` | You are assisting me with maximizing a black-box function. The function is **Description Card** [Function Patterns]. Suggest **Description Card** [Dimensionality] promising starting points in the range $[0,1]$ˆ **Description Card** [Dimensionality]. Return the points strictly in JSON format as a list of **Description Card** [Dimensionality]-dimensional vectors. Do not include any explanations, labels, formatting, or extra text. The response must be strictly valid JSON. |
| Candidate sampling `LLAMBO` | The following are past evaluations of a black-box function. The function is **Description Card** [Function Patterns]. **Data Card** The allowable ranges for x is $[0, 1]$^ **Description Card** [Dimensionality]. Recommend a new x that can achieve the function value of **Target Score** . Return only a single **Description Card** [Dimensionality]-dimensional numerical vector with the highest possible precision. Do not include any explanations, labels, formatting, or extra text. The response must be strictly valid JSON. |
| Surrogate modeling `LLAMBO` | The following are past evaluations of a black-box function, which is **Description Card** [Function Patterns]. **Data Card** The allowable ranges for x is $[0, 1]$^ **Description Card** [Dimensionality]. Predict the function value at x = $x$. Return only a single numerical value. Do not include any explanations, labels, formatting, or extra text. The response must be strictly a valid floating-point number. |
| Candidate generation `LLAMBO-light` | The following are past evaluations of a black-box function, which is **Description Card** [Function Patterns]. **Data Card** The allowable ranges for x is $[0, 1]$^ **Description Card** [Dimensionality]. Based on the past data, recommend the next point to evaluate that balances exploration and exploitation: - Exploration means selecting a point in an unexplored or less-sampled region that is far from the previously evaluated points. - Exploitation means selecting a point close to the previously high-performing evaluations. The goal is to eventually find the global maximum. Return only a single **Description Card** [Dimensionality]-dimensional numerical vector with high precision. The response must be valid JSON with no explanations, labels, or extra formatting. Do not include any explanations, labels, formatting, or extra text. |

Table 5: Prompts used across different stages of `LLAMBO` and `LLAMBO-light` in the BBO task.

The tuning objective for all models is to minimize the MSE. The search spaces for the hyperparameters are specified as follows.

RF-$4D$

- `max_depth` (Maximum depth of a tree): $[-1, 50]$ (integer; $-1$ indicates no limit)
- `min_samples_split` (Minimum samples to split an internal node): $[2, 20]$ (integer)
- `min_samples_leaf` (Minimum samples required in a leaf node): $[1, 20]$ (integer)
- `max_features` (Fraction of features to consider for best split): $[0.1, 1.0]$

SVR-$3D$

- `C` (Regularization parameter): $C \in [0.01, 1000.0]$
- `epsilon` (Epsilon in the $\epsilon$-insensitive loss): $\epsilon \in [0.0001, 1.0]$
- `gamma` (Kernel coefficient for RBF kernel): $\gamma \in [0.0001, 1.0]$

XGB-$4D$

- `max_depth` (Maximum depth of a tree): $[1, 10]$ (integer)
- `learning_rate` (Step size shrinkage): $[0.01, 0.3]$
- `subsample` (Subsample ratio of the training set): $[0.5, 1.0]$
- `colsample_bytree` (Subsample ratio of columns per tree): $[0.5, 1.0]$

**Prompts design for hyperparameter tuning task.**    The prompt settings for both `LLAMBO` and `LLAMBO-light` in the hyperparameter tuning task follow the same configuration as in the BBO task ($\alpha$ and AF), with the exception of the prompt structure. In particular, the hyperparameter tuning prompts also require both the **Description Card** and the **Data Card** to capture the relevant model specifications and historical evaluations.

Each **Description Card** specifies four key components:

- `Data Patterns`: Summarize key dataset features that help the LLM understand the task.
  1. Piston simulation function: *"The dataset models the cycle time of a piston moving within a cylinder, based on seven physical input variables including mass, surface area, pressure, and temperature."*
  2. Robot simulation function: *"The dataset models the position of a planar robotic arm consisting of four rotating joints and link lengths, computing the Euclidean distance of the arm's endpoint from the origin."*
- `Model Patterns`: Describe the predictive model being used and any fixed configurations.
- `Controllable Hyperparameters`: List the tunable hyperparameters along with their types and ranges, and this matches the controllable parameters described previously.
- `Dimensionality`: The dimensions of controllable hyperparamters.

The **Data Card** for the hyperparameter tuning task may, for instance, take the form: *(C, gamma): (0.21, 12), accuracy: 0.899; (C, gamma): (0.98, 422), mean squared error: 1.00*, where each entry reflects a past evaluation consisting of a specific hyperparameter configuration and its corresponding performance metric (i.e., MSE).

Together with the **Description Card**, which outlines the model and search space, the complete prompt structure used in both `LLAMBO` and `LLAMBO-light` is illustrated in Table 6.

| Phases | Prompts |
|---|---|
| Warmstarting
`LLAMBO`
`LLAMBO-light` | You are assisting with automated machine learning using **Description Card** [`Model Patterns`] for a regression task. **Description Card** [`Data Patterns`]. Model performance is evaluated using mean squared error. I'm exploring a subset of hyperparameters defined as **Description Card** [`Controllable Hyperparameters`].

Please suggest **Description Card** [`Dimensions`] diverse yet effective configurations to initiate a Bayesian optimization process. Return the points strictly in JSON format as a list of **Description Card** [`Dimensions`]-dimensional vectors. Do not include any explanations, labels, formatting, or extra text. |
| Candidate sampling
`LLAMBO` | The following are examples of the performance of a **Description Card** [`Model Patterns`] measured in mean square error and the corresponding model hyperparameter configurations. **Data Card** **Description Card** [`Data Patterns`] The allowable ranges for the hyperparameters are: **Description Card** [`Controllable Hyperparameters`]. Recommend a configuration that can achieve the target mean square error of **Target Score** . Return only a single **Description Card** [`Dimensions`] -dimensional numerical vector with the highest possible precision. The response needs to be a list and must be strictly valid JSON. Do not include any explanations, labels, formatting, or extra text. |
| Surrogate modeling
`LLAMBO` | The following are examples of the performance of a **Description Card** [`Model Patterns`] measured in mean square error and the corresponding model hyperparameter configurations. The model is evaluated on a regression task. **Data Card** **Description Card** [`Data Patterns`] Predict the mean square error when the model hyperparameter configurations are set to be $x$. Return only a single numerical value between 0 and 1. Do not include any explanations, labels, or extra text. The response must be strictly a valid floating-point number. |
| Candidate generation
`LLAMBO-light` | The following are examples of the performance of a **Description Card** [`Model Patterns`] measured in mean square error and the corresponding model hyperparameter configurations. **Data Card** **Description Card** [`Data Patterns`] Based on the past data, recommend the next point to evaluate that balances exploration and exploitation: - Exploration means selecting a point in an unexplored or less-sampled region that is far from the previously evaluated points. - Exploitation means selecting a point close to the previously high-performing evaluations. The goal is to eventually find the global maximum. Return only a single **Description Card** [`Dimensionality`]-dimensional numerical vector with high precision. The response must be valid JSON with no explanations, labels, or extra formatting. Do not include any explanations, labels, formatting, or extra text. |

Table 6: Prompts used across different stages of `LLAMBO` and `LLAMBO-light` in the hyperparameter tuning task.

# G    3D PRINTING DETAILS

We define the controllable design parameters of the printer via a comprehensive correlation analysis, and the selected variables of interest are summarized below.

- Nozzle Temperature: Temperature of the hot-end nozzle in °C.

- Z Hop Height: The vertical lift of the nozzle during travel (non-printing) moves.

- Coasting Volume: Volume of filament not extruded at the end of a line.

- Retraction Distance: Distance (mm) the filament is pulled back before a travel move.

- Outer Wall Wipe Distance: Distance (mm) the nozzle continues moving after the outer wall ends.

## G.1    QUALIFYING THE STRINGING PERCENTAGE

An image-based metric is used to qualify the stringing percentage. Printed parts were photographed under consistent lighting conditions against a black background. Each image was converted to grayscale to simplify processing, and a fixed region of interest (ROI) was cropped to capture the space between the two vertical columns (see the left panel of Figure 10). This region should appear empty when no stringing is present.

To differentiate potential stringing from the background, a pixel intensity threshold was selected through trial-and-error. Pixels with intensity below the threshold were set to black, while those above were set to white (see the right panel of Figure 10). The stringing percentage was then calculated as the ratio of white pixels to the total number of pixels within the ROI. This approach offers a fast and consistent approximation of stringing severity across multiple prints.

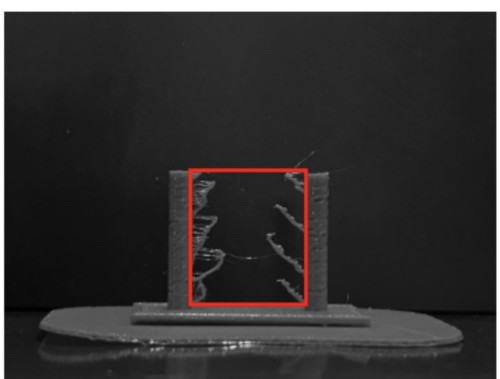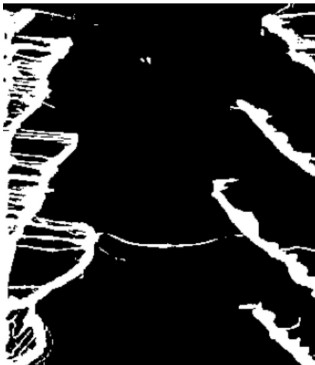

Figure 10: Grayscale image (BO, iteration 2) of the printed part with the region of interest (left panel), and white pixels approximating the stringing amount (15.9%) over the region of interest (right panel).

## G.2    PROMPTS DESIGN

The settings of LLMs are the same as in Appendix F.1. The system prompt is *You are an AI assistant that helps me optimize the 3D manufacturing process by controlling parameters.* An example of the **Data Card** is *"(Nozzle Temperature, Z Hop Height, Coasting Volume, Retraction Distance, Outer Wall Wipe Distance): (235, 0.3, 0.06, 4, 0.3), Stringing percentage: 12%.* We also need a **Parameter Description Card** to describe the controllable and fixed variables, which is *You are allowed to adjust only five slicing parameters: **Nozzle Temperature**: Range 220–260°C (step: 1°C), **Z Hop Height**: Range 0.1–1.0 mm (step: 0.1 mm), **Coasting Volume**: 0.02–0.1 mm$^3$ (step: 0.01 mm$^3$), **Retraction Distance**: 1.0–10.0 mm (step: 1 mm), and **Outer Wall Wipe Distance**: 0.0–1.0 mm (step: 0.1 mm) Slicing settings below are fixed: Retraction Speed = 60 mm/s, Travel Speed = 178 mm/s, Fan Speed = 60%. Other slicing settings are set to be the software's default values.*

The warmstarting prompt (for `LLAMBO-light` and `LLAMBO`), candidate sampling prompt (for `LLAMBO`), surrogate modeling prompt (for `LLAMBO`), and candidate generation prompt(for `LLAMBO-light`) are shown in Table 7.

| Phases | Prompts |
|---|---|
| Warmstarting `LLAMBO` `LLAMBO-light` | You are assisting with process planning for 3D printing a simple part using Overture PETG filament on an Ender 3 Pro in a room-temperature environment (around 22°C). The objective is to reduce stringing as much as possible, using knowledge of PETG printing behavior. **Parameter Description Card** After each print, stringing is measured via an image-based algorithm, returning a percentage between 0 and 100%. You must now propose 2 promising combinations of Nozzle Temperature (°C), Z Hop Height (mm), Coasting Volume (mm³), Retraction Distance (mm), Outer Wall Wipe Distance (mm) that are likely to minimize stringing, based on your understanding of PETG behavior. Format your answer strictly as a valid JSON list of 5-dimensional vectors. Each vector should be: [Nozzle Temperature (°C), Z Hop Height (mm), Coasting Volume (mm³), Retraction Distance (mm), Outer Wall Wipe Distance (mm)]. Do not include any explanations, labels, formatting, or extra text. |
| Candidate sampling `LLAMBO` | The following are past evaluations of the stringing percentage and their corresponding Nozzle Temperature (°C), Z Hop Height (mm), Coasting Volume (mm³), Retraction Distance (mm), Outer Wall Wipe Distance (mm) values: **Data Card** **Parameter Description Card** Recommend a new ([Nozzle Temperature (°C), Z Hop Height (mm), Coasting Volume (mm³), Retraction Distance (mm), Outer Wall Wipe Distance (mm)) that can achieve the stringing percentage of **Target Score**. Instructions: Return only one 5D vector: '[Nozzle Temperature (°C), Z Hop Height (mm), Coasting Volume (mm³), Retraction Distance (mm), Outer Wall Wipe Distance (mm)]'. Ensure the values respect the allowed ranges and increments. Respond with strictly valid JSON format. Do not include any explanations, comments, or extra text. |
| Surrogate modeling `LLAMBO` | The following are past evaluations of the stringing percentage and the corresponding Nozzle Temperature (°C), Z Hop Height (mm), Coasting Volume (mm³), Retraction Distance (mm), Outer Wall Wipe Distance (mm). **Data Card** **Parameter Description Card** Predict the stringing percentage at ([Nozzle Temperature, Z Hop Height, Coasting Volume, Retraction Distance, Outer Wall Wipe Distance) = x. The stringing percentage needs to be a single value between 0 to 100. Return only a single numerical value. Do not include any explanations, labels, formatting, percentage symbol, or extra text. |
| Candidate generation `LLAMBO-light` | The following are past evaluations of the stringing percentage and their corresponding Nozzle Temperature (°C), Z Hop Height (mm), Coasting Volume (mm³), Retraction Distance (mm), Outer Wall Wipe Distance (mm) values: **Data Card** **Parameter Description Card** Your goal is to recommend the next setting to evaluate that balances exploration and exploitation: Exploration favors regions that are less-sampled or farther from existing evaluations. Exploitation favors regions near previously low stringing percentages. The ultimate objective is to find the global minimum stringing percentage. The ideal stringing percentage is 0%. Instructions: Return only one 5-dimensional vector: [Nozzle Temperature (°C), Z Hop Height (mm), Coasting Volume (mm³), Retraction Distance (mm), Outer Wall Wipe Distance (mm)]. Ensure the values respect the allowed ranges and increments. Respond with strictly valid JSON format. Do not include any explanations and comments. |

Table 7: Prompts used across different stages of `LLAMBO` and `LLAMBO-light` in the 3D printing experiment.

