# OpenReview forum: "$\texttt{LLINBO}$: Trustworthy LLM-in-the-Loop Bayesian Optimization"
_ICLR.cc/2026/Conference — Submitted to ICLR 2026_

### Official Review · Reviewer_m9hu · 2025-10-29

**Soundness:** 2
**Presentation:** 2
**Contribution:** 2
**Rating:** 2
**Confidence:** 4

**Summary:**

This paper presents three different approaches on how to provide acquisition functions for Bayesian Optimisation with the use of an LLM together with a Gaussian process surrogate model. The motivation for this approach in the paper is the notion that the LLM will somehow contain "contextual" information. The first approach uses a linear combination of the LLM acquisition and the GP acquisition, the second uses a rejection criteria based on the GP surrogates acquisition function while the third uses the LLM proposal to create a constrained surrogate.

The paper ends with a set of experiments comparing the proposed method to several different baselines. The experiments are on classic BO tasks and the more interesting on a 3d printing task.

**Strengths:**

The paper is somewhat original given that the field of including LLMs into BO loops has not extensively been explored in academia. There is a substantial amount of work in this sector that is never published as a lot of the work is done in an industrial setting. While the authors are clearly open that this is by no means the "finished article" it is easy to motivate why there should be a bigger focus in this area.

**Weaknesses:**

The paper is not very clearly written and its not obvious to actually decipher what the authors propose. There is a lot of intuitive arguments and while this sadly comes with the territory when working with black-box LLMs this could have been done better. The main part which is really unclear to me is how the interaction with the LLM actually works. In the paper it is referred to as "we assume that the client can query the LLM". In the case of saying the Ackley function, what is even contextual information, what is it that the LLM actually provides and how and what does the client query?

Furthermore, in the experimental evaluation, what would have been interesting to see is how often the LLMs choices are used. Especially in the "justify" setting this is something that could be quantified.

The experimental setting is somewhat lacklustre, the basic BO examples are not really too interesting and as can be seen from the results a basic BO loop does well on these experiments. What is not clear from the paper is what the actual BO loop is, what is the surrogate model, what is the acquisition etc.

**Questions:**

- 211 :: am I correct that you use a linear combination of the LLM and the GP proposed locations? Can you explain what motivates this?
- Please provide information about how the LLM is actually queried what is the structure of this and how is the agent, and what is the agent that does this?
- Clearly describe the experimental set-up for the baselines, what is the BO loop?
- What do you mean by contextual information, can you describe what this in the example of optimising the Brain-2D or Ackley-6D? To me it is very unclear what this is making it hard to dechiper what you are proposing.

---

> ### Author Response · Authors · 2025-11-24
> **Response to Reviewer m9hu: Part 1**
>
> We sincerely thank the reviewer for the time and effort devoted to providing feedback, and for highlighting several points that could be further clarified or expanded. In **Part 1**, we address the reviewer’s main concerns regarding the prompts used for the LLM agent. In **Part 2**, we include an ablation study on the prompt variant and respond to the questions related to Line 211 (Q1) and clarify the underlying mechanism of $\texttt{LLINBO-Justify}$, as raised in the weakness section.
>
> > **Main Concern: How the LLM is actually queried.**
>
> We thank the reviewer for the question regarding how the LLM is queried. Our querying mechanism follows the emerging literature on LLM-assisted BO, particularly the state-of-the-art $\texttt{LLAMBO}$ framework of [1]. Designing prompting strategies is not our contribution. Instead, our focus is on making any LLM-assisted BO pipeline reliable via $\mathcal{GP}$-based validation and hedging. Therefore, we have deferred the complete prompt templates to Appendix~E--F. Below we also add some details for convenience (though the more comprehensive answer is provided in the appendix).
>
> At each iteration, the LLM ($\texttt{ChatGPT-3.5-turbo}$) receives two components: (i) a **Description Card** summarizing qualitative structural information about the objective function, and (ii) a **Data Card** containing past $(x,y)$ evaluations. These Description Cards provide contextual information that a $\mathcal{GP}$ surrogate cannot access. For example, for Hartmann-4$D$ we describe the function as: `A 4-dimensional, non-convex, multi-modal function composed of weighted, anisotropic Gaussian-like bumps centered at different points, making it highly non-separable and challenging to optimize.`
>
> $\texttt{LLAMBO}$ uses this information in a two-step loop: **(a)** the LLM proposes several candidate points; **(b)** the LLM is queried repeatedly to predict the function value at each candidate. Step (b) is repeated multiple times to estimate both a **mean** and an **uncertainty** for each proposed point, which $\texttt{LLAMBO}$ then uses to select the next design (e.g., via a UCB criterion).
>
> > The following are past evaluations of a 6-dimensional, non-convex, multi-modal black-box function.
> > The function exhibits a nearly flat outer region and a deep central basin, creating many local optima
> > around a single global optimum. It is symmetric and separable, but optimization remains challenging due
> > to the abundance of local maxima.
> >
> > x: [0.7211, -1.4408, 0.4220, 0.8851, -0.5934, 1.3072], f(x): 7.2369
> > x: [-1.8849, -0.6514, 1.7930, 1.3292, -1.1058, -0.2345], f(x): 9.7728
> > x: [1.0984, 0.0321, -0.7104, -0.2353, 0.8427, 1.6632], f(x): 7.8270
> > x: [0.4435, -1.2293, -0.9225, 0.1197, 1.9442, -1.1561], f(x): 9.3042
> > x: [-0.8422, 0.5174, 0.2230, -1.6914, 0.6509, -0.1133], f(x): 7.0945
> > x: [1.4277, 1.2196, -1.5629, -0.5853, -0.2402, 0.3171], f(x): 8.7127
> > x: [-0.1745, -1.9376, 1.0738, 0.7924, -1.4688, 0.5946], f(x): 9.9861
> >
> > The allowable domain for x is **[0, 1]^6**.
> > Based on the observed evaluations, recommend the next point to evaluate that **balances exploration and exploitation**:
> > – **Exploration**: select a point in an unexplored or sparsely sampled region far from previously evaluated points.
> > – **Exploitation**: select a point near previously strong-performing evaluations.
> >
> > The goal is to find the global **maximum**.
> >
> > **Return only one 6-dimensional numerical vector in valid JSON format.**
> > No explanations, no labels, no extra formatting.
>
> > **Details in BO setup.**
>
> Regarding the BO setup, we use the $\texttt{SingleTaskGP}$ model from $\texttt{BoTorch}$ as the surrogate. The $\mathcal{GP}$ uses a learned constant mean and a Matérn-5/2 kernel with automatic relevance determination.
>
> > Reference:
>
> [1] Large Language Models to Enhance Bayesian Optimization.

---

> ### Author Response · Authors · 2025-11-24
> **Response to Reviewer m9hu: Part 2**
>
> ## Ablation Study on Problem Context
>
> To fully address your concern, we have now included an ablation study examining the influence of problem context on the overall performance of our proposed algorithms. Specifically, we conducted a prompt-robustness study in the hyperparameter tuning task using three variants:
>
> - **Fully informed**: historical data + model description + tunable parameters
> - **Partially informed**: historical data + model description
> - **Minimally informed**: historical data only
>
> We report performance using the improvement proportion  $I_t = \frac{\mathrm{regret}(0) - \mathrm{regret}(T)}{\mathrm{regret}(0)},$
> where higher values indicate better performance. Each experiment is repeated ten times. Results are shown below.
>
> | **Method**                 | **Fully informed** | **Partially informed** | **Minimally informed** |
> |---------------------------|-------------------|------------------------|------------------------|
> | $\texttt{LLINBO-Transient}$   | 0.05 ± 0.01       | 0.05 ± 0.01            | 0.04 ± 0.01            |
> | $\texttt{LLINBO-Justify}$     | 0.06 ± 0.03       | 0.06 ± 0.03            | 0.06 ± 0.02            |
> | $\texttt{LLINBO-Constrained}$ | 0.06 ± 0.02       | 0.06 ± 0.01            | 0.06 ± 0.02            |
>
> The results demonstrate that $\texttt{LLINBO-Justify}$ and $\texttt{LLINBO-Constrained}$ are highly robust to prompt variation due to their performance-based acceptance mechanisms. $\texttt{LLINBO-Transient}$ may be affected during early iterations, but eventually achieves competitive performance as it gradually transitions from LLM-guided exploration to $\mathcal{GP}$-driven exploitation.
>
> ---
>
> ### (1) Motivation of Line 211
>
> We do not use a convex combination of the LLM’s and GP’s suggested points because we do not assume convexity of the underlying objective function. Instead, for our simplest approach, $\texttt{LLINBO-Transient}$, the variable $z_t$ is sampled from a Bernoulli distribution:
>
> $z_t \in \{0,1\}, \qquad \mathbb{P}(z_t = 1) = p_t,\quad  \mathbb{P}(z_t = 0) = 1 - p_t .$
>
> If $z_t = 1$, the next design $x_t$ is chosen from the GP; if $z_t = 0$, it is chosen from the LLM. This probabilistic switching mechanism adaptively selects between the two information sources.
>
> In contrast, $\texttt{LLINBO-Constrained}$ blends the two sources more structurally: when the LLM’s suggestion is promising, the GP’s posterior at that point is adjusted upward via finite sampling, producing a *constrained GP*. This mechanism leverages both the structure of the GP and the quality of the LLM’s proposal, enabling stronger coordination between the two modules.
>
> ---
>
> ### (2) Underlying mechanism of $\texttt{LLINBO-Justify}$
>
> We appreciate the reviewer’s insightful suggestion to analyze how frequently the LLM’s suggestions are accepted during optimization. From Figures 3 and 4, $\texttt{LLAMBO-light}$ performs strongly in the Levy-2$D$ black-box optimization task but less effectively in the Piston XGB-4$D$ hyperparameter tuning task. Our goal is to show that $\texttt{LLINBO-Justify}$ can:
>
> 1. **Hedge risk** by reducing reliance on the LLM when it performs poorly, and
> 2. **Benefit from LLM guidance** when it performs well.
>
> To examine this, we conducted two additional experiments using $\texttt{LLINBO-Justify}$:
>
> - Hyperparameter tuning on **Piston XGB-4$D$**
> - Black-box optimization on **Levy-2$D$**
>
> For each experiment, we record the proportion of iterations (out of 20) in which the LLM-proposed design is selected. Each experiment is repeated 10 times, and the initial dataset size equals the problem dimension.
>
> | **Task** | **Proportion of iterations using LLM suggestions** |
> |--------|-----------------------------------------------------|
> | Hyperparameter tuning (Piston XGB-4$D$) | $0.105 \pm 0.222$ |
> | Black-box optimization (Levy-2$D$) | $0.295 \pm 0.0548$ |
>
> These results show that $\texttt{LLINBO-Justify}$ effectively adapts to the LLM’s reliability: leveraging it more when beneficial (Levy-2$D$) and suppressing it when detrimental (Piston XGB-4$D$). We will include this experiment and a more detailed visualization in the next version of the paper. We kindly refer the reviewer to Appendix~C.3 for full results.

---

### Official Review · Reviewer_1aud · 2025-10-30

**Soundness:** 3
**Presentation:** 2
**Contribution:** 3
**Rating:** 6
**Confidence:** 3

**Summary:**

The paper introduces LLINBO, a hybrid framework that integrates LLMs with statistical surrogate models like Gaussian Processes (GPs) to enhance trustworthiness in black-box optimization (BBO). LLMs excel in low-data regimes by leveraging contextual knowledge for early exploration, but they lack explicit surrogate modeling, calibrated uncertainty, and transparency, leading to risks in exploration-exploitation balance and theoretical reliability. LLINBO addresses this by using LLMs for initial design proposals while progressively relying on GPs for principled exploitation as data grows. Three specific mechanisms are proposed: LLINBO-Transient (transient LLM influence that fades over iterations), LLINBO-Justify (LLM justifies proposals, GP verifies feasibility), and LLINBO-Constrained (LLM proposals constrained within GP uncertainty bounds). These draw inspiration from federated learning and come with regret-based theoretical guarantees ensuring asymptotic optimality similar to standard BO. The framework is evaluated through simulations and a real-world proof-of-concept in 3D printing, demonstrating improved efficiency and robustness.

**Strengths:**

1. Pioneers a hybrid LLM-GP approach for BO, explicitly modeling LLM degradation over time and hedging with statistical surrogates. The three variants offer flexible integration, addressing opacity and uncertainty issues in pure LLM-assisted BO.
2. Provides rigorous regret guarantees for each mechanism by extending classical BO results to the hybrid setting.
3. Balances LLM's contextual strengths (e.g., few-shot learning from prompts) with GP's uncertainty quantification. The 3D printing application shows real-world potential in costly evaluation scenarios like drug discovery or hyperparameter tuning.

**Weaknesses:**

1. Relies on GP surrogates; no discussion of alternatives like neural networks. LLM prompting (e.g., for justifications) may be sensitive, with no ablation studies mentioned. Theoretical guarantees assume smooth functions and high-probability bounds, potentially limiting generality.
2. Combining LLMs and GPs could increase computational cost (e.g., LLM queries per iteration), but no analysis of scalability or comparisons to baselines in terms of runtime.
3. Proof-of-concept in 3D printing is promising, but without full results, it's unclear how it compares to pure GP-BO or LLM-only methods across diverse benchmarks.

**Questions:**

Please see the Weaknesses.

---

> ### Author Response · Authors · 2025-11-23
> **Response to Reviewer 1aud: part 1**
>
> We sincerely thank the reviewer for recognizing the novelty of our proposed framework and the theoretical regret bounds derived for each method. In **Part 1**, we address the raised concerns, except for the two additional experiments: **(i)** the ablation study via prompt variants and **(ii)** the empirical analysis of computational time. which are presented in **Part 2**.
>
> > **W1a: Reliance on** $\mathcal{GP}$ **surrogates without discussion of alternatives (e.g., neural networks)**.
>
> Our reliance on $\mathcal{GP}$ surrogates is driven by two central reasons.
> - Most importantly, $\mathcal{GP}$s are one of the very few surrogate models for which we have strong, well-established generalization guarantees and mean-concentration bounds. These are essential for establishing the trustworthiness of our hybrid LLM-BO framework, whose main goal is precisely to ensure reliability even when the LLM becomes noisy or unreliable. Unfortunately, comparable theoretical guarantees for deep neural networks (or their Bayesian counterparts) remain highly vacuous and, in many cases, mathematically intractable. Without these guarantees, we would not be able to derive the regret bounds or safety properties that form the backbone of our contribution.
> - $\mathcal{GP}$s remain the most commonly used surrogate in BO because they provide closed-form uncertainty quantification, which is crucial for principled exploration–exploitation. In fact, recent work in human–AI collaborative BO, federated BO, and LLM-assisted BO overwhelmingly relies on $\mathcal{GP}$s for this exact reason.
>
> For these reasons, and especially because our theoretical results critically depend on $\mathcal{GP}$ concentration properties, using $\mathcal{GP}$s as the surrogate is both natural and necessary for demonstrating that our framework can make LLM-assisted BO trustworthy.
>
> - **High-probability bounds.** For BO, all known regret guarantees are based on high-probability concentration bounds rather than deterministic guarantees. This is inherent to the BO setting, where uncertainty quantification is central and the surrogate model must satisfy probabilistic confidence conditions (e.g., UCB-style bounds) in order to control regret.
>
> - **Smoothness assumption required for the theorem and high-probability bounds.** Unfortunately, without smoothness, even classical $\mathcal{GP}$-BO has no meaningful regret bounds, and extending BO theory to fully non-smooth or misspecified functions remains a very hard open problem. Our goal is to make LLM-assisted BO trustworthy under the same assumptions that make classical BO trustworthy.
>
> > **Q3: Full results for 3D printing experiments.**
>
> Due to practical constraints associated with physical experimentation, it was not feasible to evaluate all $\texttt{LLINBO}$ variants in the real 3D-printing setting. Each experimental run requires nearly a full day to produce the printed part and obtain the stringing metric, making large-scale comparisons prohibitively time- and cost-intensive. For this reason, we evaluated the most lightweight $\texttt{LLINBO}$ variant, with the goal of showing that even the simplest version of our framework can yield measurable real-world improvements under strict budget constraints. These same constraints also limited the study to seven iterations per method, which is typical in expensive-evaluation BO settings where the optimization budget is very small.
>
> Importantly, within this limited budget, $\texttt{LLINBO}$ reached near-zero stringing within just seven evaluations; substantially outperforming pure GP-BO and LLM-only baselines in this real-world task. This demonstrates that the hybrid approach is not only theoretically grounded but also practically effective in settings where evaluations are costly.
> Finally, we emphasize that the role of the 3D-printing study is to provide a proof-of-concept demonstrating that $\texttt{LLINBO}$ can translate to real physical systems. The broader performance comparison across diverse benchmarks (including pure GP-BO and LLM-only methods) is provided in our simulation studies, which are the standard way the BO community evaluates algorithmic advances. Across all benchmark functions, $\texttt{LLINBO}$ consistently shows strong and robust superiority.
>
>
> **Final Remark.** We sincerely thank the reviewer once again for the constructive and thoughtful comments. We also noticed the fair rating in the presentation category and would greatly appreciate any suggestions for improvements that we can incorporate into the revised version of the paper.

---

> ### Author Response · Authors · 2025-11-23
> **Response to Reviewer 1aud: part 2**
>
> > **W1b: Lack of ablation studies on LLM prompting.**
>
> We appreciate the reviewer’s suggestion and have now added a prompt ablation study in \textbf{Appendix C.4}. We evaluate how prompt informativeness affects performance using the \textit{Robot} dataset with a Random Forest model in a hyperparameter tuning task by varying prompt content in $\texttt{LLAMBO-Light}$ across three levels: fully informed, partially informed, and minimally informed. We report the performance using the improvement proportion $I_t = \frac{\mathrm{regret}(0) - \mathrm{regret}(T)}{\mathrm{regret}(0)},$ where higher values indicate better performance. Each experiment is repeated ten times. Results are shown in the table below.
> | **Method**                    | **Fully informed**     | **Partially informed**   | **Minimally informed**    |
> |------------------------------|------------------------|--------------------------|---------------------------|
> | $\texttt{LLINBO-Transient}$   | 0.055 ± 0.011          | 0.052 ± 0.011            | 0.040 ± 0.012             |
> | $\texttt{LLINBO-Justify}$     | 0.065 ± 0.031          | 0.059 ± 0.029            | 0.062 ± 0.029             |
> | $\texttt{LLINBO-Constrained}$ | 0.061 ± 0.021          | 0.060 ± 0.017            | 0.062 ± 0.022             |
>
> Results show that $\texttt{LLINBO-Transient}$ performs worse with less informative prompts but still converges due to the diminishing $p_t$, confirming the importance of adaptive reliance. $\texttt{LLINBO-Justify}$ remains robust and even improves under minimal information by leveraging surrogate-based justification, while $\texttt{LLINBO-Constrained}$ consistently maintains stable performance via automatic fallback to classical BO. Together, these variants highlight complementary robustness strategies, adaptive weighting, validation through surrogate models, and automatic fallback, that mitigate unreliable LLM behavior.
>
>
> > **Q2: Computational complexity of the proposed methods.**
>
> We appreciate the reviewer’s suggestion and have now included a detailed analysis in Appendix E. Here, we summarize key insights. We benchmarked the runtime of all methods on Rastrigin-2$D$ (BBO) and Robot with RF-4$D$ (HPT), using the same settings as in the main experiments, with each configuration run 10 times on a cluster with 5 nodes (dual Intel Xeon Platinum CPUs, 512 GB RAM).
>
> We summarize the results using the table below. The experiments are repeated 10 times to capture the uncertainty of the computational time.
> | **Method** | **Experiment** | **Time (s)** |
> |-----------|--------------|--------------|
> | BO | Rastrigin-2$D$ | 12.19 ± 1.12 |
> |  | Robot with RF-4$D$ | 48.76 ± 3.82 |
> | $\texttt{LLAMBO}$ | Rastrigin-2$D$ | 907 ± 12 |
> |  | Robot with RF-4$D$ | 3628 ± 48 |
> | $\texttt{LLAMBO-light}$ | Rastrigin-2$D$ | 144 ± 5 |
> |  | Robot with RF-4$D$ | 432 ± 11 |
> | $\texttt{LLINBO-Transient}$ | Rastrigin-2$D$ | 92.59 ± 11.12 |
> |  | Robot with RF-4$D$ | 278 ± 18.12 |
> | $\texttt{LLINBO-Justify}$ | Rastrigin-2$D$ | 167 ± 4 |
> |  | Robot with RF-4$D$ | 668 ± 8 |
> | $\texttt{LLINBO-Constrained}$ | Rastrigin-2$D$ | 224 ± 25.63 |
> |  | Robot with RF-4D | 896 ± 31 |
>
>
> The results align with our theoretical analysis: among the $\texttt{LLINBO}$ variants, $\texttt{LLINBO-Transient}$ is the most computationally efficient due to its diminishing reliance on the LLM (via decaying $p_t$), whereas $\texttt{LLINBO-Constrained}$ incurs the highest runtime due to repeated aggregation of retained samples, especially when $S_t$ is large. $\texttt{LLAMBO}$ is the slowest overall, even with parallelism, because it queries the LLM and updates the $\mathcal{GP}$ surrogate at each iteration, while $\texttt{LLAMBO-light}$ significantly reduces computation by avoiding repeated $\mathcal{GP}$ updates.

---

### Official Review · Reviewer_w5nX · 2025-11-01

**Soundness:** 2
**Presentation:** 3
**Contribution:** 3
**Rating:** 4
**Confidence:** 3

**Summary:**

The paper proposes LLINBO, a trustworthy hybrid Bayesian optimization framework that integrates LLMs with GP surrogates. LLMs are used during early exploration to propose contextually informed designs, while GPs gradually take over to ensure reliable uncertainty quantification and convergence guarantees. Theoretically, it derives regret bounds under RKHS assumptions. Empirically it validates the framework on synthetic benchmarks, hyperparameter tuning, and a 3D printing experiment, showing that LLINBO achieves strong early performance and near-zero stringing in real-world tests.

**Strengths:**

1.	The paper clearly motivates the need for a trustworthy hybrid. LLM assisted BO lacks explicit uncertainty and its performance degrades as data accumulate, so combining LLMs with calibrated surrogates like GPs can retain early contextual benefits and ensuring rigorous exploration–exploitation tradeoffs. The work appears to be the first to explicitly integrate LLM suggestions with GP surrogates and to analyze this interaction theoretically.
2.	Three mechanisms to coordinate LLMs and GPs during BO, including Transient (LLM exploring and shift to GP), Justify (evaluate LLM proposal against GP), and Constrained (treat LLM’s proposal as a soft constraint).
3.	The framework emphasizes trustworthy optimization. By design, LLINBO mitigates the opaque and unbounded nature of LLM suggestions, and each mechanism has a built-in safety check.
4.	Each of the three methods is accompanied by a theoretical analysis, with regret bounds developed.

**Weaknesses:**

1. The theoretical analysis treats LLM suggestions generically (e.g., by gradually ignoring them or rejecting them if the GP’s acquisition is much higher) without modeling how good the LLM’s proposals actually are. Regret bounds rely on predetermined schedules for $p_t$, $\psi_t$, or $S_t$ rather than any adaptive assessment of LLM quality. Consequently, the bounds mirror standard GP-based BO and do not show that LLM proposals improve asymptotic performance.

2. Benchmarks involve low dimensional synthetic functions and relatively small budgets (10×dimension for BBO and 5×dimension for HPT) and the real world study tests only LLINBO Transient on one 3D printing setup. It remains unclear how the approaches scale to higher dimensional or discrete design spaces, whether the constrained method is computationally feasible when many samples are needed, and how robust the results are to different LLMs or prompts.

3. LLINBO-Constrained assumes that the LLM’s suggestion $f(x_{\mathrm{LLM},t})$ is better than the current mean maximum. If this assumption is violated, the algorithm may discard the LLM proposal entirely, and the CGP sampling procedure can be computationally expensive. The practical benefit of this mechanism relative to the simpler transient or justify variants is unclear, especially since it is not tested in the real-world experiment.

4. The paper compares against a few baselines (GP-UCB, LLAMBO, and the authors’ LLAMBO-light). Given the nascent state of LLM-assisted BO, this is reasonable. One small concern is that the original LLAMBO method was modified (for practicality) – the full LLAMBO might generate multiple candidates and evaluate them with a surrogate, which could potentially yield better results than the one-shot “light” version used. Also, there are additional LLM enhanced BO methods should be compared and discussed: BioDiscoveryAgent [1], FunBO [2], LLaMEA-BO [3], and SLLMBO [4] etc. These are important works in this line of research.
---
[1] Roohani, Yusuf, et al. "Biodiscoveryagent: An ai agent for designing genetic perturbation experiments." arXiv preprint arXiv:2405.17631 (2024).

[2] Aglietti, Virginia, et al. "Funbo: Discovering acquisition functions for bayesian optimization with funsearch." arXiv preprint arXiv:2406.04824 (2024).

[3] Li, Wenhu, et al. "LLaMEA-BO: A Large Language Model Evolutionary Algorithm for Automatically Generating Bayesian Optimization Algorithms." arXiv preprint arXiv:2505.21034 (2025).

[4] Mahammadli, Kanan, and Seyda Ertekin. "Sequential large language model-based hyper-parameter optimization." arXiv preprint arXiv:2410.20302 (2024).

**Questions:**

1. In LLINBO-Constrained, the authors assume $f(x_{\mathrm{LLM},t}) > \kappa_{t-1}$. How robust is the method when this assumption is wrong? Is it possible to use a probabilistic belief about the LLM’s suggestion quality rather than a hard constraint?

2. Beyond a single candidate per iteration, have you experimented with eliciting richer information from the LLM?

---

> ### Author Response · Authors · 2025-11-23
> **Response to Reviewer w5nX: part 1**
>
> Thank you for your encouraging evaluation and insightful feedback. We sincerely appreciate the time and attention you devoted to our work. In **Part 1**, we address all concerns except W3, Q2, and the discussion for probabilistic belief about the LLM v.s. a hard constraint, which are discussed in detail in **Part 2**.
> > **W1: Lack of explicit borrowing of the LLM’s strengths.**
>
> We appreciate the insightful question. We intentionally do not externally model or quantify LLM quality because there is currently no reliable or theoretically grounded method to do so in BO: LLM outputs lack generalization guarantees, token/log-probability–based confidence is widely miscalibrated, and no Bayesian predictive formulation exists. In contrast, the GP incorporates observed evaluations and provides calibrated uncertainty with well-understood theoretical behavior, making it natural to use the GP posterior to accept, down-weight, or reject LLM proposals. Our theoretical aim is therefore to ensure **no-harm guarantees** [1]: if the LLM is unreliable, the algorithm reverts to standard BO and achieves classical regret bounds. We agree that adapting parameters (e.g., $p_t$, $\psi_t$) using measures of **LLM understanding** would be valuable, and we highlight this as promising future work.
> > W2a: **Experiments in high-dimensional settings.**
>
> We thank the reviewer for pointing out this limitation. We have added two experiments in Appendix C.2 using Levy-10$D$ and Ackley-15$D$. The results again confirm the superiority of our approach. LLM-assisted BO ($\texttt{LLAMBO}$, $\texttt{LLAMBO-light}$) performs worse than in low-dimensional cases, likely because **the prompt length becomes substantially larger in high dimensions**, making it harder for the LLM to reliably capture the underlying structure, especially as $T$ increases. In contrast, all variants of $\texttt{LLINBO}$ leverage the LLM’s few-shot capability in early iterations and rely on the $\mathcal{GP}$ for exploitation in later stages, leading to performance that surpasses all baselines.
>
> > **W2b: Only one method included in the 3D-printing.**
>
> Due to practical constraints of physical experimentation, it was infeasible to evaluate all variants in the 3D-printing setting. Each run requires ~1 day to obtain the stringing measurement, making large-scale comparisons prohibitively time- and cost-intensive. We therefore evaluated the most lightweight $\texttt{LLINBO}$ variant to demonstrate that even the simplest version of our framework can deliver measurable real-world improvements. Notably, within this strict budget, $\texttt{LLINBO}$ drove stringing essentially to zero, serving as a proof-of-concept. The broader performance comparison is reported in our simulation studies, which follow standard BO evaluation functions.
> > **W2c: Further assessment of robustness.**
>
> We thank the reviewer for this suggestion. We have added case studies in Appendix C.4 examining how different prompts (different levels of LLM knowledge) affect the proposed methods. We conducted an experiment on the HPT task (Robot with RF-4$D$) using $\texttt{LLAMBO-Light}$ under three prompt conditions:
> - **Fully informed**: historical data, data patterns, and model patterns.
> - **Partially informed**: historical data and model patterns.
> - **Minimally informed**: historical data only.
>
> Performance is evaluated using the improvement proportion from iteration 0 to $T$, where larger values indicate better results. Each experiment is repeated 10 times.
>
> |Method|Fully informed|Partially informed|Minimally informed|
> |-|-|-|-|
> |$\texttt{LLINBO-Transient}$|0.055 ± 0.011|0.052 ± 0.011|0.040 ± 0.012|
> |$\texttt{LLINBO-Justify}$|0.065 ± 0.031|0.059 ± 0.029|0.062 ± 0.029|
> |$\texttt{LLINBO-Constrained}$|0.061 ± 0.021|0.060 ± 0.017|0.062 ± 0.022|
>
> $\texttt{LLINBO-Transient}$ benefits from informative LLM prompts; when the LLM is weak, regret may rise initially, but the decaying $p_t$ ensures smooth transition to $\mathcal{GP}$-driven BO. $\texttt{LLINBO-Justify}$ is more robust, using a surrogate-based accept–reject mechanism to correct or leverage suboptimal LLM. $\texttt{LLINBO-Constrained}$ is most robust: it automatically reverts to classical BO when LLMs are uninformative.
> > **W3 \& Q1a: Assumptions and benefits of $\texttt{LLINBO-Constrained}$.**
>
> Treating an LLM suggestion as potentially superior to the current posterior mean is central to our design. Through rejection sampling, if none of the $S_t$ posterior samples at the LLM-proposed point satisfy the constraint, the method reverts to standard BO, preventing the surrogate from drifting away from evidence-supported regions. $\texttt{LLINBO-Constrained}$ requires no thresholds or schedules; only the sampling size must be chosen based on computational budget. Beyond LLM usage, constrained GP updates provide a principled mechanism for incorporating external suggestions into BO.
>
> > References:
>
> [1] Principled Bayesian Optimisation in Collaboration with Human Experts.

---

> ### Author Response · Authors · 2025-11-23
> **Response to Reviewer w5nX: part 2**
>
> > **Q1b: Probabilistic belief about the LLM v.s. a hard constraint.**
>
> In our design, the constraint is not enforced deterministically: it is evaluated by sampling from the posterior. If samples at $x_{\text{LLM},t}$ fail to satisfy the constraint, the suggestion is discarded and the method reverts to classical BO; otherwise, it is accepted and used to update the surrogate.
>
> Posterior sampling at $x_{\text{LLM},t}$ provides a probabilistic mechanism for assessing the LLM’s proposal, and the sampled value serves a second purpose: it creates an **imagined** response that enables the GP to incorporate the LLM’s signal. Without such a sampled response, one could only retain $x_{\text{LLM},t}$ based on its posterior distribution, but the surrogate would not be updated and therefore could not reflect the potential advantage of LLM. Sampling thus provides both the qualification step and a principled means to modify the surrogate when the suggestion is beneficial.
>
> > **W4a: Additional related works and benchmarks.**
>
> We thank the reviewer for highlighting additional works that incorporate LLMs into BO. We agree that leveraging LLMs within different components of BO is a promising research direction, and we will include these works in Appendix~A to provide a more complete literature overview. However, we would like to clarify that these approaches are not directly comparable as baselines due to fundamental differences in purpose and methodological design.
>
> Our framework is not intended to develop a new LLM-assisted BO algorithm perse; rather, it provides a general mechanism for making any LLM-agent–based decision process trustworthy. In principle, any existing LLM-based BO method can be plugged in as the “agent", and our framework would enhance its reliability through principled acceptance, justification, or constraint mechanisms. We focus on $\texttt{LLAMBO}$ specifically because its lightweight variant is used as the embedded LLM agent within our proposed methods, but our framework is not limited to it.
>
> In contrast, works such as LLaMEA-BO and FunBO employ LLMs to generate executable code and iteratively refine the logic. This code-generation process effectively changes the BO algorithm itself at each iteration, whereas our approach deliberately preserves the classical BO structure and focuses on integrating LLM suggestions in a principled and trustworthy way. Similarly, SLLMBO enhances the TPE surrogate using LLM signals, with the goal of proposing a new LLM-augmented optimization method, not to construct a safety framework for LLM-driven decision-making. Thus, while these works complement our motivation, they address a fundamentally different problem and therefore do not form appropriate baselines for the trustworthiness framework we develop.
> > **W4b: Concerns about integrating** $\texttt{LLAMBO-light}$ **into our experiment.**
>
> We continue to include both $\texttt{LLAMBO}$ and $\texttt{LLAMBO-light}$ as baselines in every experiment in the main paper. Their performance is very similar in both the BBO tasks and the 3D-printing experiment. In these experiments, we use $\texttt{LLAMBO-light}$ as the LLM agent primarily because of its reduced computational cost. As shown in the new empirical runtime study in Appendix E, $\texttt{LLAMBO-light}$ reduces computation time by 80% compared to $\texttt{LLAMBO}$.
>
> In the HPT, $\texttt{LLAMBO}$ occasionally outperforms $\texttt{LLAMBO-light}$. This is unsurprising, as $\texttt{LLAMBO}$ was originally designed for HPT. However, our method equipped with $\texttt{LLAMBO-light}$ still achieves competitive performance. This result further supports the effectiveness of our proposed framework: even when using a slightly simplified and weaker variant of $\texttt{LLAMBO}$, our method can still guarantee safe and reliable optimization.
>
>
> > **Q2: Eliciting richer information from LLMs.**
>
> Eliciting richer information from the LLM is part of our future research agenda, and we have added a preliminary example in Appendix C.1. We consider $\pi$BO [1], a human–AI interaction framework in which a human provides a preference function $\pi(x)$. In our adaptation, we replace the human with an LLM and automatically extract this preference.
>
> Concretely, we prompt the LLM with the historical data and the problem context, and ask it to estimate the probability that each candidate $x$ is the optimal design. These probabilities are normalized to form a valid preference distribution, which is then supplied to $\pi$BO.
>
> Empirically, whether $\pi(x)$ is fixed or updated over iterations, its performance remains consistently weaker than our proposed methods. This indicates that relying on the LLM to generate more than a single design per iteration is nontrivial, and developing systematic ways to extract richer structural information from the LLM is a promising direction for future work.
>
> > References:
>
> [2] $\pi$BO: Augmenting Acquisition Functions with User Beliefs for Bayesian Optimization.

---

### Official Review · Reviewer_nvHk · 2025-11-01

**Soundness:** 3
**Presentation:** 4
**Contribution:** 3
**Rating:** 6
**Confidence:** 4

**Summary:**

LLM in the Loop BO (LLINBO) is a hybrid framework for Bayesian optimization that combines large language models with statistical surrogate experts, with Gaussian processes as the focus in this paper. The goal is to tap the contextual reasoning strengths of LLMs while relying on principled uncertainty from statistical surrogates to enable more trustworthy optimization. The work advances the growing line of research that uses the few shot capabilities of LLMs for black box optimization, while addressing inherent limitations such as the lack of calibrated uncertainty and the opaque behavior of LLMs that makes them hard to interpret. To leverage LLMs without inheriting these risks, the authors propose a framework in which the LLM accelerates early exploration and the surrogate model increasingly guides and exploits as data accumulates.

At a high level, each BO round maintains a GP posterior, queries the LLM for a candidate $x_{LLM,t}$, evaluates that suggestion with the GP to accept, refine, or reject it (according to one of three mechanisms), and then evaluates the chosen design point. The GP “check and balance” is instantiated in three variants. LLINBO-TRANSIENT uses a Bernoulli schedule that prioritizes LLM suggestions early and shifts toward GP driven selection later. LLINBO-JUSTIFY uses the GP posterior to judge $x_{LLM,t}$, accepting it only if it lies within a specified suboptimality region; otherwise it retains the GP chosen point $x_{GP,t}$. LLINBO Constrained treats $x_{LLM,t}$ as a promising design choice and builds a constrained GP by conditioning on the event that this chosen design outperforms the current posterior mean maximizer, then acquires using a Monte Carlo approximation under the constrained posterior. Unlike the first two methods, the constrained approach avoids additional tuning parameters. All three mechanisms use GP-UCB variants s the acquisition function and come with upper bounds that ensure no regret as $T \to \infty$. The paper presents a range of empirical results to demonstrate early gains from LLM guidance and competitive or improved performance as the GP takes over for several problems.

**Strengths:**

1. The paper is well motivated and tackles a growing area of research in the BO community.

2. The authors present a clear, coherent narrative of their proposed methods.

3. The paper provides solid theoretical results that establish the no regret behavior of the mechanisms.

4. The work includes a practical demonstration that shows how LLINBO performs in a real setting.

5. The experimental section is extensive, with clearly documented details that support reproducibility.

**Weaknesses:**

1. The novelty is quite moderate. Although the third variant (LLINBO Constrained) presents a meaningful new idea, its scalability and the compute trade offs are not clearly justified with profiling or ablations.

2. Related to the above, the paper claims LLINBO is the first hybrid framework that integrates LLMs and GPs for BO. This does not seem entirely correct: for example, Kristiadi et al. (ICML 2024), “A Sober Look at LLMs for Materials Discovery,” used an LLM as a feature extractor into a GP surrogate for BO in materials discovery. The authors should clarify the novelty relative to such setups.

3. Several controlling parameters are required across the three mechanisms. Although recommended values are provided, there is no elegant or reliable procedure for selecting them across domains.

4. The synthetic functions for the black-box optimization (BBO) task do not mimic the higher dimensional settings common in BO papers as the chosen functions range from 2D to 6D. Also for the BBO tasks, LLINBO variants and standard BO are not clearly differentiated in final performance, aside from early stage gains that do not always change the end result. The authors, however, do demonstrate stronger performance on real world tasks and hyperparameter tuning settings.

**Questions:**

1. Can the authors clarify the distinction from Kristiadi et al. paper as noted earlier, and whether the claim of being the first hybrid framework combining GPs and LLMs for BO is accurate? Fundamentally, the methods are different, but the “first” claim may not be entirely correct. A clarification would be greatly welcomed.

2. Can the authors comment on the restriction of the synthetic functions to just 2 to 6 dimensions and, if possible, report performance on higher dimensional synthetic problems as typically done in BO papers (for example, 2D to 16D for low to moderate dimensional settings)? If this is not feasible, a brief justification would be helpful.

3. Can the authors discuss the cost implications across the three methods, beyond the interpretability argument that motivated the choice of LLINBO-TRANSIENT for the 3D-printing problem?

---

> ### Author Response · Authors · 2025-11-23
> **Response to Reviewer nvHK: part 1**
>
> We sincerely thank you for your encouraging evaluation and insightful comments. We truly appreciate the time you invested in engaging deeply with our work. To match the depth of your feedback, we organize our response into two parts. In the first part, we address your key concerns regarding **(1)** dimensionality, and **(2)** methodological novelty. In the second part, we provide a detailed discussion of **(1)** selecting hyperparameters (W3), **(2)** performance compared with BO (W4) and **(3)** Computational complexity (Q3).
>
> > **W1 \& W4a \& Q2: Experiment in high dimension setting, computational trade off and test of robustness via prompt varying.**
>
> To fully address your concern, we have now conducted 3 additional experiments in Appendices C.2 (high dimensional experiments in 10 and 15-dimensional simulation function.), E (an experiment on $\texttt{LLINBO-Constrained}$ with different settings of $S_1$ (initial sample size) and $S_t$ (the sample size at time $t$)), and C.4 (An experiment on hyperparameter tuning (HPT) with three different levels of contexts provided to the LLM agent). We kindly invite you to check to those appendices for more details. Our key findings are:
>
> - **High-dimensional settings.** In high-dimensional tasks, we observe that existing, purely LLM-assisted BO struggles to focus its search near the optimal region, as prompt length and contextual complexity grow substantially with dimension, leading to increasingly noisy suggestions. In contrast, our approach which uses the $\mathcal{GP}$’s calibrated uncertainty results in markedly more stable progress and **improved convergence behavior as dimensionality increases**.
>
> - **Ablation on LLM informedness.** We evaluate the proposed methods under three different levels of LLM informativeness in the HPT task. $\texttt{LLINBO-Constrained}$ emerges as the most robust method, primarily due to its accept/reject mechanism driven by posterior sampling, which protects against poor LLM suggestions. Although $\texttt{LLINBO-Transient}$ degrades when the LLM is weakly informed, because it places higher trust on the LLM in the initial iteration, its performance improves as it gradually shifts emphasis toward the $\mathcal{GP}$’s recommendations, allowing it to still move toward the optimal design.
>
> - **Computational complexity vs. performance for** $\texttt{LLINBO-Constrained}$. Appendix E includes a detailed analysis of the trade-off between computational complexity and performance. When the decay rate of $S_t$ is fixed, a larger initial value $S_0$ retains more samples during the early iterations, which is beneficial because LLM suggestions tend to be strong at this stage. Conversely, if $S_1$ is too small, the method effectively reduces to classical BO and fails to leverage any advantage from the LLM. Based on this analysis (further discussed in Appendix E), we recommend choosing $S_1$ as large as computational resources permit.
>
> > **W2 & Q1: Relationship between the proposed method and the paper A Sober Look at LLMs for Materials Discovery** [1].
>
> We would like to clarify that the two settings differ fundamentally. Our framework follows the increasingly popular LLM-agent paradigm in BO, where the LLM proposes a single next design point based on contextual information and accumulated data. Crucially, this preserves the classical BO pipeline: the $\mathcal{GP}$ remains the statistical surrogate that directly models the response function with calibrated uncertainty, guiding acquisition and maintaining the theoretical foundations of $\mathcal{GP}$-based BO.
>
> In contrast, [1] uses the LLM in a fundamentally different role. Their method does not use the LLM to propose designs; instead, it extracts features that replace the $\mathcal{GP}$’s original input space (Algorithm 2 in [1]). The $\mathcal{GP}$ is then trained on LLM-derived embeddings rather than design variables, which alters its behavior and removes the standard interpretation of the $\mathcal{GP}$ posterior as a calibrated belief over the objective function. As a consequence, this feature-engineering approach does not maintain the classical BO structure, and to the best of our understanding, makes BO-style guarantees infeasible because the $\mathcal{GP}$ no longer models uncertainty over the function being optimized.
>
> We acknowledge that [1] also aims to leverage LLMs within BO; however, this does not contradict our claim. In BO, the $\mathcal{GP}$ is used to model the belief over the unknown objective based on historical data. Our framework preserves this role: the $\mathcal{GP}$ is the judge, accepting, refining, or rejecting the LLM’s proposals according to its posterior. The framework in [1] instead changes the representation of the input space, thereby removing the $\mathcal{GP}$ from its classical position in the BO loop, and thus cannot be viewed as a variant of the proposed LLINBO paradigm.
>
> > Reference:
>
> [1] A Sober Look at LLMs for Materials Discovery.

---

> ### Author Response · Authors · 2025-11-23
> **Response to Reviewer nvHK: part 2**
>
> > **W3: Procedure for selecting hyperparameters.**
>
> We now provide the parameter selection procedure in Appendix D. Our choices are guided by **theoretical considerations** and the **empirical observation that LLM suggestions degrade over time**. For $\texttt{LLINBO-Justify}$, we set $p_t = 1 - 1/t^2$ to gradually reduce reliance on potentially unreliable LLM, reflecting that LLMs perform well mainly in familiar domains (e.g., standard HPT). For $\texttt{LLINBO-Constrained}$, LLM quality is less critical due to its data-driven acceptance mechanism. We therefore recommend a conservative decreasing schedule $\psi_t = \mathcal{O}(1/t)$, with $\psi_1$ chosen based on the acquisition function (e.g., posterior variance under UCB).
>
> Following up on the reviewer’s comment regarding $\texttt{LLINBO-Constrained}$, we emphasize that this variant is deliberately designed to require **no tuning parameters**. It operates without hyperparameter adjustment; the only user-specified quantity is the finite sampling size, determined as a simple function of the available budget. Beyond LLM usage, we view $\texttt{LLINBO-Constrained}$ as a novel and theoretically grounded mechanism for incorporating LLMs into BO.
>
> One may optionally adjust the decay of $S_t$ to hedge against unreliable suggestions. In our experiments, we set $S_1 = 10^4$ to reflect the initial computational budget and decrease it at rate $1/t^2$, based on the empirical behavior of the LLM. This allows early iterations to benefit from informative proposals, while later iterations naturally revert toward the $\mathbf{GP}$-driven modeling as the LLM’s utility declines.
> > **W4b: the performance of the proposed methods and BO.**
>
> We observe that the initial guidance provided by the LLM agent plays a substantial role in the HPT task and the 3D-printing experiment, often leading to noticeably better performance than classical BO. This is likely because the LLM can leverage much richer contextual information in these settings, such as details about the ML models being tuned or the experimental setup, which enables more informed early-stage proposals. In contrast, the context available in black box optimization (BBO) tasks is inherently limited, typically consisting only of descriptions of the function’s shape, components, or qualitative behavior. Although the LLM still improves early performance in BBO, the advantage is less pronounced. Consequently, as the optimization progresses, the performance of BO and $\texttt{LLINBO}$ tends to converge. This follows naturally from the $\texttt{LLINBO}$ design: when LLM suggestions are not accepted, either due to the probability schedule in \texttt{LLINBO-Transient} or performance-based rejection in \texttt{LLINBO-Justify} and $\texttt{LLINBO-Constrained}$, the algorithm effectively reduces to classical BO.
> > **Q3a: When to choose which variant.**
>
> While all three methods perform competitively, we recommend selection based on the practical needs (see Appendix D):
> - $\texttt{LLINBO-Transient}$ is the simplest and most interpretable. It gradually reduces reliance on LLM through a time-decaying schedule, and is appealing for users who value transparency, simplicity, or scenarios where LLM access is costly.
> - $\texttt{LLINBO-Justify}$ accepts LLM suggestions only when justified by data; preserves interpretability without modifying the BO pipeline.
> - $\texttt{LLINBO-Constrained}$ is the most robust and theoretically principled. It incorporates LLM suggestions through a probabilistic constraint that automatically hedges against uncertainty. This makes it particularly suitable for safety-critical or resource-limited settings.
> > **Q3b: Rationale for using** $\texttt{LLINBO-Transient}$ **in the application.**
>
> Due to real-world experimental constraints, each run required ~1 day to obtain the stringing metric. This prevented testing all variants and limited us to 7 iterations per method. We therefore chose the simplest variant to demonstrate that meaningful improvement is achievable even with the most lightweight approach.
> > **Q3c: Discuss the cost implications across the three methods.**
>
> We thank the reviewer for raising this concern. In response, we have added a dedicated section in Appendix E that analyzes the computational complexity of the three proposed methods in detail. Specifically, we prove that $\texttt{LLINBO-Transient}$ requires only a sublinear number of LLM queries, which is desirable given that LLM calls dominate runtime. From an algorithmic perspective, $\texttt{LLINBO-Constrained}$ incurs the highest cost due to finite-sampling aggregation, while $\texttt{LLINBO-Transient}$ is the most efficient. Runtime experiments on Rastrigin-2$D$ and Robot with RF-4$D$ confirm that $\texttt{LLINBO-Transient}$ is fastest, $\texttt{LLINBO-Constrained}$ slower, and original $\texttt{LLAMBO}$ remains slowest overall, even with parallelization, whereas $\texttt{LLAMBO-light}$ gains efficiency by avoiding repeated sampling from LLMs.

---

### Author Response · Authors · 2025-11-23
**Key changes in the revision**

We thank all reviewers for their constructive feedback. In response, we have added additional materials in the appendix to address the reviewers' suggestions, which we found highly supportive. We summarize the key changes below and sincerely invite all reviewers to refer to the revised version for more details.


- Two **high-dimensional experiments** were conducted and are presented in **Appendix~C.2** in response to comments from reviewers **nvHK** and **w5nX**. We evaluate the proposed methods on the 10-dimensional Levy function and the 15-dimensional Ackley function, and benchmark them against standard BO, $\texttt{LLAMBO}$, and $\texttt{LLAMBO-light}$. The results again confirm the advantageous proprieties of our methods. Interestingly, we find that as the dimensionality increases, and the prompt length grows accordingly, it becomes increasingly difficult for the LLM to produce effective designs as the history expands. As a result, the need to calibrate LLM proposals, as enabled by our methods, becomes even more critical.


- We now **assess the ability of** $\texttt{LLINBO-Justify}$ **to selectively accept or reject LLM-generated suggestions** in **Appendix~C.3**, addressing reviewer **m9hu**'s feedback. Specifically, we evaluate its behavior on two contrasting tasks: the 2$D$ Levy function (a black-box optimization (BBO) task where the LLM agent performs well) and the 4$D$ Piston simulation with an XGBoost surrogate (a hyperparameter tuning task where the LLM performs poorly). Using the same $\psi_t$ settings, the method accepted approximately 30\% of LLM suggestions in the former, but only 10\% in the latter. This demonstrates the effectiveness of the accept/reject scheme in automatically hedging against suboptimal LLM suggestions. However, thanks to its explicit transition from LLM to $\mathcal{GP}$, the performance is also comparable to the other two $\texttt{LLINBO}$ variants.

- An **ablation study on the proposed methods under varying prompt informativeness** is now included in **Appendix~C.4**, addressing the concerns raised by reviewers **nvHK**, **w5nX**, and **1aud**. We evaluate the robustness of the methods by providing three levels of information to the LLM. Our findings indicate that $\texttt{LLINBO-Constrained}$ is the most stable across prompt variants, likely due to its automatic hedging strategy via posterior sampling. In contrast, $\texttt{LLINBO-Transient}$ is more sensitive to prompt quality, as it places greater trust in the LLM without directly evaluating its suggestions. That said, our proposed methods remain advantageous compared to benchmarks.

- In **Appendix~E**, we provide **comprehensive materials related to computational complexity**, including **(a).** a detailed analysis of each method’s complexity (as suggested by reviewers **w5nX** and **1aud**), **(b).** empirical runtime experiments (as suggested by reviewer **1aud**), and **(c).** an evaluation of the performance–runtime trade-off for the $\texttt{LLINBO-Constrained}$ variant in response to reviewer **nvHK**. We show that $\texttt{LLINBO-Transient}$ only requires a sublinear number of LLM queries, and we derive the complexity order for all methods, identifying $\texttt{LLINBO-Constrained}$ and $\texttt{LLINBO-Transient}$ as the highest and lowest in computational complexity, respectively. Our findings suggest that initializing $\texttt{LLINBO-Constrained}$ with a larger $S_1$ (number of LLM-suggested samples) and decreasing it over time based on the LLM’s performance strikes an effective balance between efficiency and accuracy; given the diminishing reliability of the LLM in later iterations, we recommend setting $S_t = \mathcal{O}(1/t)$ (or faster decay) to reflect this behavior.

---

### Meta-Review · Area_Chair_R7dd · 2025-12-28

**Summary:**

This paper addresses a hybrid framework that integrates large language models with statistical surrogate models to optimize black-box functions. LLMs can leverage contextual knowledge to suggest promising design points in low-data regimes but they lack calibrated uncertainty which statistical surrogate models yield. This paper presents three practical strategies (LLINBO-transient, LLINBO-justify, LLINBO-transient) to combine the best of both worlds (LLMs and GPs) and show that all three methods achieve no-regret performance (asymptotically matching standard BO), effectively offering a no-harm guarantee even if the LLM provides poor suggestions. The paper is well motivated and tackles a growing area of research in the BO community. However, there are few critical concerns which should be taken into account.
1. The novelty is quite moderate. Although the third variant (LLINBO Constrained) presents a meaningful new idea, its scalability and the compute trade offs are not clearly justified with profiling or ablations. Authors have done extra experiments for ablation study during the rebuttal period, to defend this concern partially. I agree that the novelty is limited.
2. This paper presents the theoretical worst-case cumulative regret bounds for the proposed LLINBO methods are asymptotically equivalent to the standard GP-UCB bound, which is referred to as "no-harm guarantee".  While this ensures safety, it demonstrates that the theoretical analysis does not quantify any improvement from the LLM integration. The benefit of the LLM remains purely heuristic and empirical, as the theory treats the LLM as a black box without characterizing when or why it accelerates convergence.
3. A few issues on experiments (see reviewers' comments) were pointed out, most of which were addressed by the authors.

In general, the paper is easy to read and well written, presenting practical ways of leveraging LLMs with GP for BO. The empirical success depends heavily on the LLM's ability to parse specific textual descriptions of the problem. The paper needs thorough stress-test scenarios where the context is misleading or adversarial, which is critical for a paper claiming safety and trustworthiness. The no-harm guarantee protects regret, but it does not protect the wasted budget spent querying the LLM before the algorithm decides to ignore it. I hope authors found the review comments informative and can improve their paper by addressing these carefully in future submissions.

**Reviewer Concerns:**

The authors have done a nice job addressing most of the reviewers' concerns.

**Reviewer Scores:**

I expect most reviewers will maintain their original score.

---

### Decision · Program_Chairs · 2026-01-26

Reject